# Targeted Treatment of Sarcomas by Single Protein Encapsulated Doxorubicin with Undetectable Cardiotoxicity and Superior Efficacy

**DOI:** 10.3390/cancers17050881

**Published:** 2025-03-04

**Authors:** Changjun Yu, Faqing Huang, Leslie Wang, Mengmeng Liu, Warren A. Chow, Xiang Ling, Fengzhi Li, Galen Cook-Wiens, Linrong Li, Xiaojiang Cui

**Affiliations:** 1Division of Chemistry and Chemical Engineering, California Institute of Technology, 1200 E California Blvd, Pasadena, CA 91125, USA; 2Sunstate Biosciences LLC, 118 S. Berkeley Ave, Pasadena, CA 91107, USA; leslie.wang@sunstatebiosciences.com (L.W.); mml@sunstatebiosciences.com (M.L.); 3Department of Chemistry and Biochemistry, The University of Southern Mississippi, Hattiesburg, MS 39406, USA; 4Division of Hematology and Oncology, Department of Medicine, UCI Health, Orange, CA 92868, USA; wachow@hs.uci.edu; 5Department of Pharmacology and Therapeutics, Roswell Park Comprehensive Cancer Center, Elm and Carlton Streets, Buffalo, NY 14263, USA; xiang.ling@roswellpark.org (X.L.); fengzhi.li@roswellpark.org (F.L.); 6Canget BioTekpharma LLC, 701 Ellicott Street, Buffalo, NY 14203, USA; 7Department of Biomedical Sciences, Cedars Sinai Medical Center, Los Angeles, CA 90048, USA; galen.cook-wiens@cshs.org; 8Department of Breast Surgery, Peking Union Medical College Hospital, Chinese Academy of Medical Sciences and Peking Union Medical College, Beijing 100006, China; lilinrong2018@126.com; 9Department of Surgery, Samuel Oschin Comprehensive Cancer Institute, Cedars Sinai Medical Center, Los Angeles, CA 90048, USA

**Keywords:** single protein encapsulation, doxorubicin, SPEDOX-6, cardiotoxicity, toxicokinetic, soft tissue sarcoma, Ewing sarcoma, triple negative breast cancer, FcRn expression level

## Abstract

Doxorubicin (DOX) is one of the main options for treating soft tissue sarcomas. However, a cumulative lifetime limit of <450 mg/m^2^ DOX has restrained its applications. Various formulation strategies have been developed to increase DOX’s efficacy and reduce toxicities but with limited success. We recently developed single-protein-encapsulated DOX, SPEDOX-6, a novel approach for improving DOX’s efficacy and reducing its side effects. Rat model studies demonstrated superior toxicokinetic properties and undetectable cardiotoxicity for SPEDOX-6 at high doses. Since the neonatal Fc receptor (FcRn) plays an essential role in maintaining the long half-life of IgG and human serum albumin (HSA), we also investigated the relationship between SPEDOX-6’s efficacy and FcRn level. In mouse models, SPEDOX-6 remarkably suppresses HT-1080 (lowest FcRn), compared to DOX, and is significantly better than Doxil/DOX in inhibiting SK-ES-1 (highest FcRn). Combined with a study on MB-MDA-231 (medium FcRn), SPEDOX-6’s antitumor efficacy displays an inverse relationship with FcRn levels in three tumor mouse models, thereby providing a potential mechanism for SPEDOX-6 to effectively target tumors with low FcRn levels.

## 1. Introduction

As rare tumors, sarcomas represent ~0.8% of all new cancer cases in the United States, with 76% arising in soft tissue and the rest originating from the bone [1,2,3,4,5,6,7,8]. Soft tissue sarcomas (STS) are a group of heterogeneous malignancies with more than 100 histologically and biologically distinct subtypes of cancer [6]. Without distinctive symptoms from the early-stage STS, its early diagnosis is often difficult, and STS is frequently diagnosed only at an advanced stage [4,6,7]. In addition, relative to other cancers, STS occurs more frequently in young adults and adolescents [3], further aggravating the devastation caused by the disease. The survival rates for STS patients are low and have not changed for more than four decades [4,6].

Although there are several FDA-approved therapies for STS, doxorubicin (DOX) has remained the standard treatment for the past 45 years [4,6]. As a widely used anticancer drug, DOX is unfortunately associated with severe toxicities including cardiomyopathy and myelosuppression. A cumulative lifetime limit of less than 450 mg/m^2^ of DOX treatment for a cancer patient is recommended [9]. While various combinations of DOX with other agents, such as ifosfamide, dacarbazine, olaratumab, have been investigated, they have failed to demonstrate a survival benefit compared with DOX alone [4,7]. These failed clinical trials reaffirmed the efficacy of frontline DOX for patients with advanced STS and supported DOX as the core component in developing new and more efficacious STS therapeutics. Various formulation strategies based on nano-systems such as liposomes, protein assembly, and polymer conjugation have been developed to increase DOX’s efficacy and reduce toxicities but with limited success [10]. In particular, Doxil as a nanoparticle (NP) drug showed no improved clinical benefits over DOX in terms of objective response, overall survival, and progression-free survival rates [11], but displayed specific side toxicities compared with DOX, such as hand–foot syndrome, hypersensitivity reaction, stomatitis, and esophagitis [12,13,14]. In addition to DOX, developing other anthracycline derivatives continues to be an active area of research in cancer therapeutics [15,16,17]. While artificially assembled nano-systems for drug delivery bring hope for better cancer therapeutics with both high anticancer efficacy and low side effect toxicities, the translation of principles and animal studies into effective treatments that ultimately benefit cancer patients has been difficult and faces various challenges [18,19,20,21,22]. Therefore, better anticancer drugs for STS patients with higher efficacy and lower systematic toxicity are urgently needed.

We recently developed Single-Protein Encapsulation (SPE) technology to formulate and deliver various anticancer drugs [10,23,24,25,26]. The current study specifically focuses on SPEDOX-6 (Figure 1, containing nine DOX molecules per human serum albumin (HSA) molecule), which has been granted orphan drug designation by the FDA for human phase Ib/IIa clinical trials to treat STS. Although HAS–DOX interactions have been studied extensively [27,28,29,30,31,32,33,34] and different forms of HAS–DOX NPs have been prepared and studied [35,36,37,38], SPEDOX-6 is distinguished by its uniform size (around 7–9 nm, very similar to HSA), monodispersity, and desired binding strength [10], and therefore represent the first-of-its-kind HAS–DOX nanodrugs. In addition, since SPEDOX-6 contains a native HSA molecule, there is little risk of immunogenic reactions, a common concern when developing new therapeutics. Furthermore, HSA has a long circulatory half-life through the mechanism of neonatal Fc receptor (FcRn)-mediated recycling [39,40]. When HSA is endocytosed into the cell, FcRn binds to HSA in the endosome under acidic conditions and transports it back out of the cell. At neutral pH, HSA is then released into circulation. In the absence of FcRn, endocytosed HSA would be degraded by proteases in the lysosome and/or cytosol. Therefore, FcRn is a determining factor for HSA circulatory half-life and distribution. The HSA in SPEDOX-6 is in its native form and SPEDOX-6 is expected to have a long circulatory half-life in humans. In addition, SPEDOX-6 may be used to target cancer cells that express low levels of FcRn via the mechanism of increased intracellular accumulation of SPEDOX due to less efficient SPEDOX-6 recycling relative to normal cells. This FcRn-based principle of cancer targeting by drug-carrying HSA has been demonstrated on pancreatic cancer cell lines and their derived xenografts with a DOX–HSA conjugate via an acid-sensitive linker [41].

In previous studies with mouse models [10], we have reported that SPEDOX-6 displayed desirable pharmacokinetics (PK), a multifold enhancement of maximum tolerated dose (MTD), greater antitumor efficacy in the TNBC animal model (MDA-MB-231), and multifold lower free DOX concentrations in mouse heart tissues, which indirectly implicated lower cardiotoxicity compared to DOX. In another study [23], we utilized human induced pluripotent-stem-cell-derived cardiomyocytes (hiPSC-CMs), endothelial cells (hiPSC-ECs), cardiac fibroblasts (hiPSC-CFs), multi-lineage cardiac spheroids (hiPSC-CSs), patient-specific hiPSCs, and multiple human cancer cell lines to compare the anticancer efficacy and reduced cardiotoxicity of SPEDOX-6 relative to DOX. Cell viability assays and immunostaining in human cancer cells, hiPSC-ECs, and hiPSC-CFs revealed robust uptake of SPEDOX-6 and efficacy in killing these proliferative cell types. In contrast, hiPSC-CMs and hiPSC-CSs exhibited substantially lower cytotoxicity during SPEDOX-6 treatment compared to DOX. SPEDOX-6-treated hiPSC-CMs and hiPSC-CSs maintained their functionality, demonstrating the potential of SPEDOX-6 to alleviate cardiotoxic side effects associated with DOX while maintaining its anticancer potency [23]. These unique properties of SPEDOX-6 are highly desirable for cancer therapeutics and provide the basis for the FDA’s approval of orphan drug designation and granting of human Ib/IIa clinical trials.

In the present study, we aim to achieve four objectives: (1) directly confirming GLP-grade SPEDOX6’s reduced cardiotoxicity at multifold MTD in an SD rat model using microscopic examination of the fixed rat heart tissues; (2) demonstrating the superb toxicokinetics (TK) of GLP-grade SPEDOX-6 in Sprague–Dawley (SD) rat model; (3) demonstrating the superior antitumor efficacy of SPEDOX-6 against two commonly used human sarcoma cell lines, HT-1080 and SK-ES-1, compared to DOX and Doxil; and (4) investigating whether SPEDOX-6’s antitumor efficacy is correlated to FcRn expression level in STS, TNBC and Ewing Sarcoma (ES) mouse models. Our results demonstrate that SPEDOX-6 not only shows undetectable cardiotoxicity at multifold MTD and superior TK properties in SD rat models, but SPEDOX-6 is also highly effective against xenograft sarcomas in mice derived from HT-1080 and SK-ES-1 cell lines without increased toxicity relative to that of DOX and Doxil. HT-1080 was originally isolated from connective tissue as a fibrosarcoma cell line for the STS model. However, it was recently found to have an R132C mutation in IDH1, which is a characteristic commonly associated with chondrosarcomas. As a result, HT-1080 was reclassified as dedifferentiated chondrosarcoma [42]. Therefore, HT-1080 may be used as a model for both STS and bone sarcomas. Furthermore, the combined data from our studies strongly suggest that the antitumor efficacy of SPEDOX-6 may be inversely correlated with FcRn expression level in cancer cells, as expected based on the FcRn-mediated HSA recycling mechanism. Therefore, SPEDOX-6 may become an effective cancer-targeting therapeutic for sarcomas and other cancers that express low levels of FcRn [41].

## 2. Materials and Methods

### 2.1. Material and Instruments

HSA (25% solution, lot #: K125A6871) and DOX hydrochloride (DMF #: 16178, batch #: DR020421) were purchased from Octapharma USA (San Diego, CA, USA) and Gemini PharmChem (Mannheim, Germany), respectively. Methanol, ethanol, and other chemicals and supplies were purchased from VWR. UV spectrum measurement and quantitation were conducted on a UV-1600 PC spectrometer (VWR), Radnor, PA, USA. Non-GLP-grade SPEDOX-6 (Figure 1) was prepared and characterized according to the published methods [10]. GLP-grade SPEDOX-6 (Lot #: R1059-01-091) and GMP-grade SPEDOX-6 (lot #: 23SD015) were manufactured by Societal CDMO, San Diego, LLC at 6828 Nancy Ridge Drive, Suite 100, San Diego, CA, USA.

### 2.2. GLP Toxicology Study

“Non-GLP dosing range study of SPEDOX-6” and “GLP-toxicology study of SPEDOC-6” were conducted in JOINN Laboratories Inc at 2600 Hilltop Dr, BLDG C, Richmond, CA, USA, an FDA-certified GLP Lab via the fee-paid services. Animal care was compliant with the SOPs of JOINN Laboratories, which is fully accredited by the Association for Assessment and Accreditation of Laboratory Animal Care International. Procedures used in both studies were approved by the Institutional Animal Care and Use Committee (IACUC) with serial number S-ACU22-0824.

#### 2.2.1. Non-GLP Dosing Range Study of SPEDOX-6

The objective of this study was to evaluate the acute toxicity of SPEDOX-6 after a single dose via intravenous infusion to SD rats followed by a 14-day observation period. Based on the body weight, a total of 50 SD rats (25/sex) were randomly assigned to 5 groups with 5/sex/group in groups 1 to 5. Rats in group 1 were treated with 0.9% saline injection as the negative control group (0 mg/kg), and rats in groups 2 to 5 were treated with non-GLP-grade SPEDOX-6 at doses of 10, 20, 30, and 40 mg/kg, respectively. The treatments were administered via intravenous (IV) infusion (tail vein) for a single dose. The dose volume was 10 mL/kg and the dose rate was set at 3.33 mL/kg/min. The first dosing day was defined as Day 1. Animals were observed at least 4 h after dosing. Parameters evaluated in this study included clinical observations, body weight (BW) change, and food consumption. All surviving animals in groups 1 to 5 were euthanized on Day 15 and had a complete necropsy examination, organ weighing, and macroscopic examination. During the study, neither mortality nor moribundity was noted in animals in the negative control group or the 10, 20, and 30 mg/kg groups. However, one female animal in the 40 mg/kg group was found dead on Day 12.

#### 2.2.2. GLP Toxicology Study of SPEDOX-6

The objectives of this study were to evaluate toxicity, toxic target organs, and TK profile of GLP-grade SPEDOX-6 administered by IV to SD rats once every 3 weeks for 4 consecutive weeks (2 doses in total), and to evaluate the reversibility of toxicity following a 3-week recovery period. A comparative study was carried out with the positive control article (DOX) to provide animal study data for the follow-up study of the test article.

Based on body weight, a total of 214 SD rats (107 rats/sex) were randomly assigned to 10 groups with 15/sex/group in groups 1, 2, 4, and 5 for the toxicity study, 10 rats/sex in group 3 for the toxicity study, 5 rats/sex in group 6 for the TK study, and 8 rats/sex/group in groups 7, 8, 9, and 10 for the TK study. The rats in groups 1 and 6 were administered with 0.9% saline as the negative control groups (0 mg/kg); the rats in groups 2 and 7 were administered with DOX as the positive control groups (3.5 mg/kg); the rats in groups 3 and 8, groups 4 and 9, and groups 5 and 10 were administered with 7.5, 15, and 25 mg/kg of SPEDOX-6 as the low, middle, and high dose groups, respectively. The animals were administered by IV infusion via tail vein once every three weeks for 4 consecutive weeks, with 2 doses in total, and had a 3-week recovery period. The dose volume was 10 mL/kg and the dosing speed was set at 3.33 mL/kg/min. The first dosing day was defined as Day 1.

Parameters evaluated in this study included those of clinical observations (including injection site observation), BW change, food consumption change, ophthalmoscopic examinations, hematology, coagulation, clinical chemistry, urinalysis, and TK. The first 10 rats/sex/group in groups 1 to 5 were euthanized one week after the last dosing (on Day 29), and the remaining 5 rats/sex/group were euthanized on Day 50 following a 3-week recovery period. All animals in groups 1 to 5 were subjected to a complete necropsy examination, organ weighing, and macroscopic examination. Histopathological evaluation was performed on the animals in groups 1 and 5.

TK study: After IV infusion of 0.9% saline, DOX, and SPEDOX-6 once every three weeks for 4 consecutive weeks, blood samples were collected at 0.083 h, 0.5 h, 1 h, 2 h, 4 h, 8 h, 24 h, and 48 h after the start of dosing on Day 1 and Day 22. All collected blood samples were processed and the total DOX concentrations in rat plasma were determined by validated LC-MS/MS methods. TK parameters were analyzed using non-compartmental analysis (NCA) with Win Nonlin to assess the systemic exposure of SPEDOX-6 or DOX in SD rats.

Cardiotoxicity: All heart slides from groups 1, 2, 4, and 5 were processed at JOINN Laboratories S using routine histological methods (embedded in paraffin, sectioned, mounted on slides, and stained with hematoxylin and eosin (H&E), etc.). The proportion and severity of myofibrillar loss and vacuolization were evaluated under a light microscope and scored according to the criteria in Table 1. Myofibrillar loss and vacuolization in the heart were evaluated for all animals at terminal and recovery necropsies under a light microscope and scored according to the semi-quantitative scoring criteria for cardiotoxicity.

### 2.3. In Vivo Animal Model Study

Both HT-1080 and SK-ES-1 mouse model studies were performed at Roswell Park Comprehensive Cancer Center Animal Facility following the animal protocol approved by the Institutional Animal Care and Use Committee (IACUC).

#### 2.3.1. HT-1080 Mouse Model with Non-GLP-Grade SPEDOX-6

HT-1080 cell line (CCL-121) was purchased from ATCC. After growing in Eagle’s Minimum Essential, HT-1080 cells were harvested by trypsinization and washed twice with PBS. HT-1080 cells (1 × 10^6^ per injection) were suspended in 200 µL of a 1:1 solution of ice-cold PBS and Matrigel (Corning Incorporated, Corning, NY, USA) solution. HT-1080 cancer xenograft tumors were first generated by injecting 1 × 10^6^ cancer cells into the flank area of severe combined immunodeficiency (SCID) mice (CB17SC, strain C.B-*Igh-1^b^*/IcrTac-*Prkdc^scid^*, Roswell internal breeding). After the tumors grew to 800–1200 mm^3^, they were isolated, and approximately 50 mg of non-necrotic tumor mass was subcutaneously implanted into the flank area of individual mice. Almost equal numbers of female (16 mice) and male (17 mice) mice were used in this experiment. When the implanted xenograft tumors grew to 250 to 350 mm^3^ at 7 days after tumor transplantation, mice were randomly divided into 8 groups for intravenous injection: (1) vehicle (saline, 4 females), (2) DOX (5 mg/kg, 4 females), (3) SPEDOX-6 A (non-GLP-grade) (15 mg/kg, 4 females), (4) SPEDOX-6 B (non-GLP-grade) (17.5 mg/kg, 5 females), (5) vehicle (saline, 4 males), (6) DOX (5 mg/kg, 4 males), (7) SPEDOX-6 A (non-GLP-grade) (15 mg/kg, 5 males), and (8) SPEDOX-6 B (non-GLP-grade) (17.5 mg/kg, 5 males). The intended schedule for drug or vehicle treatment was weekly for 3 doses. However, mice in groups 2 and 6 with DOX at 5 mg/kg lost >20% BW after 2 doses, indicating severe toxicity. As a result, the third dose was canceled. Mice in group 1 on Day 9 and in group 5 on Day 6 were sacrificed due to the large tumor size with diameter ≥20 mm. Tumor volume (TV) and BW were measured two to three times per week or daily depending on the condition of the mouse. TV was calculated using the formula: v = 0.5 (L × W^2^). Progression at the endpoint was a tumor size with diameter ≥20 mm or a moribund condition.

#### 2.3.2. SK-ES-1 Mouse Model with GLP-Grade SPEDOX-6

SK-ES-1 cell line (HTB-86) was purchased from ATCC. After growing in Eagle’s Minimum Essential, SK-ES-1 cells were harvested by trypsinization and washed twice with PBS. SK-ES-1 cells (1 × 10^6^ per injection) were suspended in 200 µL of a 1:1 solution of ice-cold PBS and Matrigel (Corning Incorporated, Corning, NY, USA) solution. SK-ES-1 cancer xenograft tumors were first generated by injecting 1 × 10^6^ cancer cells into the flank area of SCID mice (CB17SC, strain C.B-*Igh-1^b^*/IcrTac-*Prkdc^scid^*, Roswell internal breeding). After the tumors grew to 800–1200 mm^3^, they were isolated, and approximately 50 mg of non-necrotic tumor mass was subcutaneously implanted into the flank area of individual mice. An equal number of female (16 mice) and male (16 mice) mice were used in this experiment. When the implanted xenograft tumors grew to 250 to 350 mm^3^ at 7 days after tumor transplantation, mice were randomly divided into 8 groups for intravenous injection: (1) vehicle (saline, 4 females + 4 males), (2) DOX (3.5 mg/kg, 4 females + 4 males), (3) Doxil (4 mg/kg, 4 females + 4 males), (4) SPEDOX-6 (GLP-grade) (30 mg/kg, 4 females + 4 males). The dosing schedule is qwk × 3 (weekly for 3 doses). Mice in groups 1 and 2 on Day 9 were sacrificed due to the large tumor size with diameter ≥20 mm. TV and BW were measured two to three times per week or daily depending on the condition of the mouse. TV was calculated using the formula: v = 0.5 (L × W^2^). Progression at the endpoint was a tumor size with diameter ≥20 mm or a moribund condition.

### 2.4. Immunohistochemistry (IHC) Analysis on Ki67 and Cleaved Caspase-3

Deparaffinized tissue sections were rehydrated and incubated in 1 × pH6 citrate buffer (Invitrogen at Waltham, MA, USA, Cat #00–5000) for 20 min using a DAKO PT Link. With an Autostainer, the following steps and reagents were used for IHC analysis: (1) Incubation in 3% H_2_O_2_ for 15 min; (2) incubation with 10% normal goat serum for 10 min (Thermo Fisher, Waltham, MA, USA, #50062Z); (3) incubation with Avidin/Biotin block for 10 min (Vector Labs, Newark, CA, USA Cat# SP-2001); (4) incubation with primary Ki67 antibody (Abcam, Cambridge, UK, #ab15580 or cleaved caspase-3 (Asp175) antibody (Cell Signaling, Danvers, MA, USA, Cat #9661) diluted in 1% BSA for 30 min; (5) incubation with secondary goat anti-rabbit (Vector Labs #BA-1000) for 15 min; (6) incubation with ABC reagent (Vector Labs Cat #PK 6100) for 30 min; (7) incubation with DAB substrate (Dako, Glostrup, Denmark, Cat #K3467) for 5 min; (8) counterstained with DAKO hematoxylin for 20 s; (9) coverslipped slides.

### 2.5. Statistical Analysis

#### 2.5.1. HT-1080 Mouse Model Study

A linear mixed model was used for TV on Day 6 and Day 20, controlling for Day 0 TV, group, day, and group-by-day interaction and gender. A similar model was used for BW percentage but did not control for Day 0 since the percentage is calculated from Day 0. A compound symmetric covariance was used to model the correlation in repeated measures. Post-hoc pairwise comparisons between groups on each day were made using a Tukey–Kramer adjustment for multiple testing. Detailed statistic results for HT-1080 are listed in Appendix A.

#### 2.5.2. SK-ES-1 Mouse Model Study

In order to compare TV changes under different treatments at each time point, two-way ANOVA and mixed-effects analysis were performed using GraphPad Prism 9.5 for Mac OS (GraphPad Software, Boston, MA, USA, www.graphpad.com (accessed on 18 February 2025)), when appropriate. The xenograft model experiments consisted of 3 to 4 replicates per condition, the results of which were presented as mean ±SEM (standard error of the mean) in figures. *p* < 0.05 was considered statistically significant. To evaluate treatment efficacy, the tumor growth inhibition ratio (TGI, %) was calculated using the following formula: TGI = (1 − (mean TV of treated group)/(mean TV of control group)) × 100%. To evaluate toxicity, normalized BW was calculated using the following formula: normalized BW = (BW)/(BW on Day 0) × 100%. Two-way ANOVA and mixed-effects analysis were used to calculate the TV and normalized BW changes in mice under different treatments at each time point, when appropriate. In particular, the Kruskal–Wallis test and ordinary one-way ANOVA were used to calculate the normalized BW changes in mice under SPEDOX-6 treatment, when appropriate. The analyses were performed using GraphPad Prism 9.5. The xenograft model experiments consisted of 3 to 4 replicates per condition, the results of which were presented as mean ± standard deviation in figures. *p* < 0.05 was considered statistically significant.

## 3. Results

Our previous studies with mouse models, human cancer cell lines, and induced pluripotent stem cell-derived cell lines have established that SPEDOX-6 possesses desirable PK, robust uptake by proliferative cell types, great antitumor efficacy, and reduced side effects, compared with DOX. These results suggest that SPEDOX-6 has great potential to become a promising new cancer therapeutic with improved efficacy and reduced side effect toxicity. The current study is part of IND (investigational new drug)-enabling studies, designed to evaluate the general toxicity and cardiotoxicity of SPEDOX-6 in rodent animal models in order to obtain approval for SPEDOX-6’s IND applications. The experimental results described in the following sections have satisfied the FDA to grant permission to conduct human Ib/IIa clinical trials of SPEDOX-6 on STS.

### 3.1. Evaluation of Acute Toxicity of Non-GLP-Grade SPEDOX-6 in SD Rat Model

The acute toxicity of non-GLP-grade (lab-prepared) SPEDOX-6 was evaluated in SD rats via IV route in order to obtain dose ranges of SPEDOX-6 that will be used to design dose levels in the upcoming GLP toxicology. IV infusion of SPEDOX-6 to SD rats with a single dose at 10, 20, 30, and 40 mg/kg resulted in a decrease in BW, food consumption (Appendix A), and organ mass (data not shown due to a large amount of data of different rates, doses, and organs). The degree of reduction in BW/food consumption/organ mass was inversely correlated with the drug dose for SPEDOX-6. From the data, MTD and the severely toxic dose in 10% (STD10) were determined to be 30 and 40 mg/kg, respectively. In comparison to DOX with LD10 (lethal dose in 10%) at 7.4 mg/kg (SD rat model) [43], non-GLP-grade SPEDOX-6 reduces DOX’s toxicity by more than 5-fold.

### 3.2. GLP-Toxicology Study

Due to the severe immuno-reactions to HSA by non-rodent models, such as pigs, dogs, and rabbits, the proposed GLP toxicology of SPEDOX-6 using only SD rats was approved by the U.S. FDA and conducted accordingly. Three dose levels of SPEDOX-6, low (7.5 mg/kg), medium (15 mg/kg), and high (25 mg/kg), along with a positive control (DOX at 3.5 mg/kg) have been chosen for this study based on the above acute toxicity study.

TK on SPEDOX-6: Rat serum DOX concentration–time profiles are shown in Appendix A. Key TK parameters of DOX and SPEDOX-6 in each group after the first (Day 1) and last (Day 22) treatments are shown in Table 2 and Table 3, respectively.

The results from the TK studies may be summarized as follows. (1) After IV infusion of DOX at 3.5 mg/kg and GLP-grade SPEDOX-6 at 7.5 (low dose), 15 (middle dose), and 25 mg/kg (high dose) (<MTD at 30 mg/kg from the above acute toxicity using non-GLP-grade SPEDOX-6) into SD rats, DOX and GLP-grade SPEDOX-6 were detectable in plasma samples on the first and last dosing days (D1 and D22). (2) For both DOX at 3.5 mg/kg and SPEDOX-6 at the three different doses, there was no significant gender difference in the systemic exposure (AUC_last_) of DOX and SPEDOX-6 of each dose group. (3) After repeated administration on Day 22, there was some systemic accumulation of DOX and SPEDOX-6 in all SD rats, with apparent gender differences. (4) The systemic exposure of SPEDOX-6 increased with the increasing dose, and the rate of increase in exposure was greater than the rate of increase in dose. By comparing SPEDOX-6 to DOX at three dose ratios of 2.14 (i.e., 7.5/3.5), 4.29 (i.e., 15/3.5), and 7.14 (i.e., 25/3.5), the corresponding C_max_ ratios (C_max_ ratio/dose ratio) (in red), AUC_last_ ratios, and (AUC_last_ ratio/dose ratio) ratios (in green) were calculated and are shown in Table 4, revealing a quantitative assessment on enhancement of the total exposure for SPEDOX-6 vs DOX (positive control) in the same amount, from 7.24 to 17.16 times. Therefore, the total exposure (AUC_last_) of SPEDOX-6 at three doses was significantly higher than DOX at the equivalent dose for both male and female rats. Based on the AUC_last_ ratio/dose ratio, the FDA requested and granted the initial dose of SPEDOX-6 at 20 mg/m^2^ for the first-in-human testing, instead of the standard DOX dose of 75 mg/m^2^.

Cardiotoxicity Study: All dissected heart tissues were microscopically examined and scored according to the semi-quantitative scoring criteria described in Table 1. No myofibrillar loss and vacuolization were observed in heart tissues. The score of all groups was 0 based on the scoring criteria, indicating undetectable cardiotoxicity for doses up to 2 × 25 mg/kg (total doses of 50 mg/kg) of SPEDOX-6. In comparison, the literature reported that a single high dose of DOX at 5–10 mg/kg in male SD rats induced cardiotoxicity [44].

BW change and food consumption study: SD rats’ BW and food consumption–time profiles for GLP-grade SPEDOX-6 are shown in Appendix A, respectively. Over 7 weeks, rat BW change and food consumption showed a dose-dependent decrease relative to the negative control. SPEDOX-6 at 15 mg/kg showed no greater decrease than the positive control 3.5 mg/kg DOX (at MTD). Male rats are more sensitive to DOX and GLP-grade SPEDOX-6 compared to female rats, especially at a dose of 3.5 mg/kg DOX and a dose of 25 mg/kg GLP-grade SPEDOX-6. In addition, 25 mg/kg SPEDOX-6 showed higher toxicity in male rats than 3.5 mg/kg DOX. Within the male groups, there is a clear order, for BW loss, SPEDOX-6 at 25 mg/kg > DOX at 3.5 mg/kg > SPEDOX-6 at 15 mg/kg > SPEDOX-6 at 7.5 mg/kg; for food consumption in reverse order, SPEDOX-6 at 25 mg/kg < DOX at 3.5 mg/kg < SPEDOX-6 at 15 mg/kg < SPEDOX-6 at 7.5 mg/kg. However, four female groups only showed slight differences among them in the first 4 weeks and did not display any difference in ≥5 weeks for both BW change and food consumption. Based on the results, 25 mg/kg SPEDOX-6 has a similar toxicity profile to that of 3.5 mg/kg DOX (MTD) in female rats, with a 7.1-fold increase in MTD. For male rats, SPEDOX-6’s MTD is estimated between 15 and 25 mg/kg, a 4.2–7.1-fold increase over that of DOX.

### 3.3. Evaluation of Antitumor Efficacy for Non-GLP-Grade SPEDOX-6 Against HT-1080

Given that DOX is the standard therapy for STS as the first-line treatment, we tested the effect of SPEDOX-6 on STS tumor growth using the established HT-1080 xenograft STS mouse model (SCID mice) expressing the lowest level of FcRn (<2 TPM) among cell lines [45]. Of note, although HT-1080 was originally isolated from connective tissue as an STS model, it was recently reclassified as a dedifferentiated chondrosarcoma line due to characteristic IDH1 mutation [42]. To improve DOX therapy, it is often highly desirable to deliver near its MTD to cancer cells while limiting the drug’s toxicity. Consequently, our tumor efficacy experiments were designed to test SPEDOX-6 doses at multiple MTDs of DOX based on our previous studies [10]. Since STS happens to both males and females, we also explored possible gender preference/bias for DOX and SPEDOX-6 by using both male and female mice. In the first attempt at an in vivo STS xenograft model study, we tried non-GLP-grade SPEDOX-6 doses at 15 and 20 mg/kg, with the latter being SPEDOX-6’s MTD based on our previous TNBC model studies (nude BALB/c mice) [10]. However, all mice in the 20 mg/kg treatment group lost more than 20% BW and subsequently went into a moribund state, indicating exceeding the MTD. The experiments were terminated following the euthanization of the mice. The second experimental design was modified to include the vehicle control, DOX at an established MTD dose of 5 mg/kg (DOX group) [43,44], and SPEDOX-6 at 15 and 17.5 mg/kg (SPEDOX-6 A and B groups) of DOX equivalent with weekly tail vein injections 3 times.

#### 3.3.1. Antitumor Efficacy and Toxicity Evaluations

Due to the fast growth rate of HT-1080, all male mice in the vehicle control group had to be euthanized on Day 6 (Figure 2A). For the DOX group, only two doses were administered due to severe BW loss and the scheduled third dose was canceled. On Day 6, the tumor size of all three treatment groups was significantly smaller compared to the control group; while both DOX and SPEDOX-6 A groups showed similar slow tumor growth from Day 0, in contrast, the tumor in the SPEDOX-6 B group shrunk by 29%. From Day 6 to 13, the tumor size of all three groups continued to shrink. However, the tumor in the DOX group started to grow from Day 13 to 20. On the contrary, the SPEDOX-6 A and B groups continued the trend of tumor shrinkage until Day 17, when the SPEDOX-6 A group slightly reversed the trend until Day 20. On the other hand, the SPEDOX-6 B group reached 90% tumor reduction on Day 17 and maintained a similar tumor size until Day 20. As shown in Figure 2B, over the 20-day period, the DOX group had a 38% increase in TV, while the SPEDOX-6 A and B groups shrunk tumor size by 35% and 86%, respectively. In stark contrast, the tumor of the control group grew rapidly by 241% in just 6 days, when the male mice had to be terminated due to the large tumor size (Table 5 and Figure 2A).

The toxicity of the treatment agents was evaluated by BW change over time, which is an established method for early-stage preclinical studies [10,46,47]. As shown in Figure 2C, the BW of all mice decreased over the treatment period due to the xenograft tumor burden, and the BW decrease in SPEDOX-6 A and B groups was slightly less than the DOX group, although statistically insignificant. Therefore, SPEDOX-6 at the dose of 17.5 mg/kg × 3 did not show higher toxicity than DOX at the dose of 5 mg/kg × 2, indicating SPEDOX-6’s MTD (DOX-equivalent) is >5 times that of DOX. The effects of SPEDOX-6 and DOX on tumor growth inhibition and BW changes are summarized in Table 5.

#### 3.3.2. Treatment Effects in Male and Female Mice

We further explored whether there is a gender bias in the treatment effects of SPEDOX-6 on HT-1080. For the control groups, tumors in males grew faster than in females (Appendix A). All the males and females in the control groups were sacrificed on Day 6 and 9, respectively, due to fast tumor growth. Additionally, one male in the DOX group was euthanized on Day 12 due to severe BW loss. TV of the male control group increased by 272% on Day 6, compared to a 221% increase for the female control group on Day 9. For the DOX group, the tumor size changed over 20 days by −11% and 74% in the male and female groups, respectively. In comparison, male and female groups treated with 15 mg/kg of SPEDOX-6 showed TV changes of −73% (partial remission, defined as >50% TV reduction) and 24%, respectively, from Day 0 to 20. Remarkably, 17.5 mg/kg of SPEDOX-6 treatment over the same period led to 89% and 84% TV reduction (partial remission) in the male and female groups, respectively. Furthermore, two males in the SPEDOX-6 A group on Day 9 and 16 and one male in the SPEDOX-6 B group on Day 13 reached a tumor-free state (complete remission). Examination of individual mouse responses to SPEDOX-6 treatment reveals great differences between genders (male vs female) and between doses (15 vs. 17.5 mg/kg). At 15 mg/kg dose (SPEDOX-6 A), two out of five males attained partial and complete remission, while only one of four females achieved partial remission. With 17.5 mg/kg (SPEDOX-6 B) treatment, four out of five and one out of five males attained partial and complete remission, respectively, and all five females reached partial remission status. However, the variation in treatment response among individuals of both genders is relatively large, leading to high statistical deviations (ns, *p* > 0.05) both within and between gender groups.

BW change in response to treatment was similar in male and female groups (Appendix A). In both gender groups, DOX caused slightly more BW loss over time than SPEDOX-6, indicating that both male and female mice experienced no higher side effect toxicity from multi-fold DOX-equivalent of SPEDOX-6 compared with DOX. However, the difference is statistically insignificant among the three treatment groups in both males and females (Table 5).

Photographic images of tumors removed at the end of experiments for each treatment group are shown in Figure 3. These images indicate that SPEDOX-6 at both treatment doses of 15 and 17.5 mg/kg achieved better antitumor effects than DOX at its MTD dose (5 mg/kg). In addition, 17.5 mg/kg of SPEDOX-6 uniformly reduced the tumor size by over 80% with high statistical significance. Therefore, TV–treatment time curves, partial/complete remission status, and photographic tumor images all demonstrate that SPEDOX-6 at a 17.5 mg/kg DOX-equivalent dose is very effective in suppressing HT-1080 tumor growth without displaying higher systemic toxicity compared to DOX. Furthermore, SPEDOX-6 appears more effective against tumors in males than in females, although statistical analysis indicated an insignificant difference (ns, *p* > 0.05).

### 3.4. Evaluation of Antitumor Efficacy for GLP-Grade SPEDOX-6 Against SK-ES-1 (ES Model)

In the above section, HT-1080 (STS mouse model) with the lowest FcRn expression level among cell lines was evaluated using non-GLP-grade SPEDOX-6, showing great antitumor efficacy. To explore whether FcRn level plays a role in SPEDOX-6’s antitumor efficacy, we conducted a similar study on a mouse model with the highest FcRn expression level among cell lines, SK-ES-1 (ES) at 894 TPM [45]. Since ES is also treated by DOX as the first-line option [48], our experimental designs include four groups—the control, DOX, Doxil (liposome DOX nanoparticles), and GLP-grade SPEDOX-6 using both male and female SCID mice. In the above HT-1080 xenograft SCID mouse model study, the intended three weekly doses of DOX at 5 mg/kg had to be reduced to two weekly doses due to toxicity. Therefore, DOX’s dose was adjusted to 3.5 mg/kg for three weekly injections. In addition, Doxil was dosed at 4.0 mg/kg (MTD) and GLP-grade SPEDOX-6 was dosed at 30 mg/kg (almost 2× that of non-GLP-grade SPEDOX-6) of DOX equivalent with weekly tail vein injections three times.

#### 3.4.1. Antitumor Efficacy and Toxicity Evaluations

Due to the fast growth rate of SK-ES-1 and the ineffectiveness of DOX at its MTD (3.5 mg/kg) against tumor growth, all mice in the control group and DOX group had to be euthanized on Day 10 (Figure 4A). On Day 8, DOX reached a TGI of 18.4%, while Doxil had a TGI of 36.4%. SPEDOX-6 showed a much higher TGI of 54.0%, which is significantly better than the control group (****, *p* < 0.0001). Furthermore, SPEDOX-6 reduced tumor volume compared to the DOX group (*, *p* = 0.0183). On Day 21, SPEDOX-6 greatly reduced TV compared to the Doxil group (**, *p* = 0.0016). As shown in Figure 4A and Table 6, over the 10-day period, the control group and the DOX group had a 293% and 377% increase in TV, respectively, indicative of the ineffectiveness of DOX for inhibiting SK-ES-1 (ES). Over a 21-day period, TV increases of 569% (Doxil) and 307% (SPEDOX-6) were observed (Table 6). Although SPEDOX-6 is significantly better than Doxil at inhibiting SK-ES-1, SPEDOX-6 is relatively less sensitive to SK-ES-1 (ES) compared to efficacy in suppressing HT-1080 (STS).

The toxicity of the treatment agents was evaluated by BW change over time, as shown in Figure 4B and Table 6. BW of all mice did not change more than 20% over the testing period, though the GLP-grade SPEDOX-6 group had an initial decrease in BW, but it recovered almost completely. Therefore, SPEDOX-6 at the dose of 30 mg/kg did not show higher toxicity than DOX at the dose of 3.5 mg/kg, indicating GLP-grade SPEDOX-6’s MTD (DOX-equivalent) is >8 times that of DOX. By carefully analyzing antitumor efficacy and toxicity data for four study groups in males and females, no gender bias in the mouse model with SK-ES-1 was observed, which is different from the above HT-1080 mouse model study.

#### 3.4.2. Immunohistochemical (IHC) Study

The paraffin-embedded tumor tissue sections were subjected to H&E staining (tissue morphology), Ki67 staining (a cellular marker for proliferation), and cleaved/active caspase-3 staining (a marker for programmed cell death). H&E, Ki67, and cleaved/active caspase-3 staining on one tissue from the vehicle group, DOX group, Doxil group, and SPEDOX-6 treatment group are shown in Figure 5A. By visualizing Figure 5A, two left panels, the vehicle, and DOX treatment group tumor tissues seem to similarly display high cancer cell density with large nuclei and high Ki67 but low cleaved caspase-3 levels. In the third panel from the left, the Doxil treatment group’s tumor tissue seems to show slightly reduced cancer cell density and lower Ki67 but slightly higher cleaved caspase-3 levels relative to the left panel, indicating the limited antitumor effect by Doxil. On the right panel, the SPEDOX-6 treatment group tissue seems to show reduced cancer cell density and lower Ki67 but higher cleaved caspase-3 levels relative to the Doxil treatment panel, indicating better antitumor efficacy by SPEDOX-6 than Doxil, which is consistent with the tumor TV study. Furthermore, Ki67 positive and caspase-3 active cells in all four samples in Figure 5A were scored, as shown in Figure 5B,C. They clearly indicate that all three drug treatments reduced Ki67-positive cells, with SPEDOX-6 achieving the highest effect. The difference between Doxil and SPEDOX-6 is significant (***, *p* = 0.001) (Figure 5B). For caspase-3, DOX does not show a statistical difference from the control. While both Doxil and SPEDOX-6 activated caspase-3, the activation by SPEDOX-6 was more than 2 times higher than that by Doxil. Caspase-3 activation by SPEDOX-6 is significantly different from the other three samples (****, *p* < 0.0001) (Figure 5C).

Taken together, TV and IHC staining/scoring results convincingly demonstrate that SPEDOX-6 at a dose of 30 mg/kg is effective in suppressing SK-ES-1 (ES) tumor growth compared to DOX at 3.5 mg/kg and Doxil at 4.0 mg/kg. However, DOX at its MTD (3.5 mg/kg) is ineffective in suppressing SK-ES-1 tumor growth. Furthermore, Doxil at 4.0 mg/kg (MTD) had limited antitumor efficacy, only slightly better than DOX but much less effective than SPEDOX-6.

The results from the above studies can be summarized as follows. (1) MTD and STD10 of non-GLP-grade SPEDOX-6 in SD rat model was 30 mg/kg and 40 mg/kg, respectively, which reduce DOX’s toxicity by more than 5-fold; (2) the total exposure for SPEDOX-6 in SD rat model is enhanced 7–17 times depending on dosage, relative to DOX; (3) SD rat cardiotoxicity at a high dose (50 mg/kg) of GLP-grade SPEDOX-6 was undetectable, while a single injection of high-dose DOX at 5–10 mg/kg led to observable cardiotoxicity [44]; (4) SPEDOX-6 of both 15 and 17.5 mg/kg at cumulative doses of 45 and 52.5 mg/kg was well tolerated by mice and no more toxic than DOX at its MTD of 10 mg/kg cumulative dose; (5) SPEDOX-6 at both doses achieved better antitumor efficacy than DOX, and 17.5 mg/kg of SPEDOX-6 remarkably suppressed HT-1080 tumor growth with statistical significance while imposing no severe toxicity, with 3 out of 10 mice reaching tumor-free status; (6) GLP-grade SPEDOX-6 is significantly better than DOX and Doxil at their respective MTD in inhibiting SK-ES-1 tumor growth; (7) SPEDOX-6 has less efficacy against SK-ES-1 (ES) than HT-1080 (STS); (8) GLP-grade SPEDOX-6 enhances DOX’s MTD by more than 8 times in SK-ES-1 mouse model.

### 3.5. FcRn Expression Level and SPEDOX-6’s Antitumor Efficacy

Since HSA’s circulation and distribution are regulated by FcRn and SPEDOX-6 contains native HSA in its monomeric form as the carrier for DOX, we would like to know whether SPEDOX-6’s antitumor efficacy is influenced by FcRn expression level, and if so, what kind of correlation exists between efficacy and FcRn expression levels. To address these questions, we need to (1) have different tumor models treated by SPEDOX-6, (2) have a consistent parameter to measure antitumor efficacy, and (3) consider the intrinsic DOX sensitivity of different tumor models. TGI measures the TV difference between a drug-treated mouse group and the control mouse group (no treatment), and it is a good parameter for drug efficacy. However, the tumor in the control group grows fast and often needs to be sacrificed well before the end of experiments. As a result, TGI is only available for early time points before the tumor in the control group gets to the allowable size limit. On the other hand, TV change (%) between the initial and final time points might be a good parameter when comparing different drug treatments.

We previously used non-GLP-grade SPEDOX-6 to treat MDA-MD-231 (TNBC, 18 TPM FcRn) using the nude BALB/c mouse strain [10]. The current study adds two more tumor models: HT-1080 (STS) and SK-ES-1 (ES). Relevant data for these studies, including DOX IC_50_ [49] and FcRn expression level [45], are listed in Table 7. Three sets of TV vs treatment time profiles are shown in Figure 6A. It is obvious that SPEDOX-6 at 17.5 mg/kg in the STS mouse model is more effective than 20 and 30 mg/kg in the TNBC and ES mouse models, respectively.

Assuming that mice strains and SPEDOX-6 grades have only small effects on antitumor efficacy, we can test whether DOX IC_50_ values or FcRn levels have any correlation to SPEDOX-6’s antitumor efficacy against these three types of cancers. As shown in Figure 6B, DOX IC_50_ for STS is 3 and 27 times less than that for ES and TNBC, respectively. However, SPEDOX-6’s antitumor efficacy follows the decreasing order of STS > TNBC > ES. Therefore, there is no obvious correlation between DOX IC_50_ values and SPEDOX-6’s antitumor efficacy. However, when antitumor efficacy is plotted against the FcRn level of different tumor cells (Figure 6C), a clear correlation can be seen. Furthermore, plotting TV% change vs log (FcRn value) displays an apparent inverse linear relationship (Figure 6D).

## 4. Discussion

As a widely used and effective anticancer drug, DOX is always associated with severe well-known toxicities including cardiomyopathy and myelosuppression. As recommended, a cumulative lifetime limit of <450 mg/m^2^ for a cancer patient [9] restricts its clinical applications. Furthermore, after the approval of DOX by the FDA for over 40 years, the mechanisms of DOX’s cardiotoxicities (acute and chronic) are still debatable. As described in our previous publications [10,23], SPEDOX-6 has shown a low amount of free DOX in mouse heart tissues and low cytotoxicity to hiPSC-CMs and hiPSC-CSs while maintaining its antitumor efficacy in human cancer cells. This GLP toxicology study provides direct evidence that cardiotoxicity of SD rats for both genders after IV infusion of two high doses at 25 mg/kg (total cumulative dose of 50 mg/kg) with two cycles and a 3-week recovery period (total 50 days) is microscopically undetectable. However, a single injection of 5–10 mg/kg for DOX would have caused severe cardiotoxicity for male SD rats [44]. Recently, we used an HPLC-equipped size exclusion column (SEC) to analyze all components of GMP-batch SPEDOX-6 (Lot #: 23SD015), indicating <0.5% of the free DOX (small molecules) and >99.5% of the encapsulated DOX (large complexes). Therefore, the low amount of free DOX in SPEDOX-6 can explain the undetectable cardiotoxicity in SD rats.

As described in the previous section, SPEDOX-6 has substantially enhanced the total exposure (AUC_last_) at three dose levels, ranging from 7.24- to 17.16-fold enhancement, compared to DOX at equivalent doses for both genders. However, the t_1/2_ of HSA in rats and humans are 14.8 and 450 h, respectively [50]. Accordingly, we expect SPEDOX-6 would circulate in humans much longer than in rats. Based on the AUC_last_ ratio/dose ratio, the FDA requested and granted the initial dose of SPEDOX-6 at 20 mg/m^2^ for the first-in-human testing in the approved phase Ib/IIa protocol, instead of the standard DOX dose of 75 mg/m^2^.

Due to SPEDOX-6’s dramatic reduction in toxicity, undetectable cardiotoxicity, and longer circulation time (TK study) in rat model, we proposed the highest dose at 310 mg/m^2^ for GMP grade SPEDOX-6 for a total of six cycles in the current human escalation trial. It is reasonable to expect that SPEDOX-6 can achieve 310 mg/m^2^ and a total amount of SPEDOX-6 after six cycles can reach 1860 mg/m^2^, exceeding the cumulative lifetime limit of 450 mg/m^2^ [9] by more than four times, which is unprecedented.

While the antitumor efficacy–FcRn relationship (Figure 6D) is derived from only three tumor mouse model studies and needs to be confirmed from additional tumor models, the relationship may be explained by the HSA cellular recycling mechanism via FcRn [39,40]. FcRn binds to endocytosed HSA at acidic pH of the endosome and brings it back to the cell surface, releasing HSA at neutral pH and returning it to the circulation system. Since SPEDOX-6 is a monomeric native HSA with embedded DOX molecules [10], the same HSA recycling mechanism would apply to SPEDOX-6. Low concentration of FcRn reduces SPEDOX-6 recycling, leading to increased SPEDOX-6 accumulation in the tumor cell. Subsequent DOX dissociation from HSA and/or HSA enzymatic hydrolysis generates free DOX, which diffuses to the cell nucleus to interfere with essential replication and transcription processes, thereby achieving antitumor effects. Accordingly, SPEDOX-6 recycling efficiency in xenograft tumor cells would be inversely correlated with the FcRn expression level, which is consistent with the antitumor efficacy–FcRn relationship in Figure 6D. The relationship will be tested in SPEDOX-6’s current clinical trials on STS, where the archival tumor tissues of STS patients will be analyzed for FcRn expression level in order to correlate to SPEDOX-6 treatment efficacy, as requested by the FDA. If the correlation can be established in STS patients, future pre-screen tests on FcRn level may be developed for SPEDOX-6’s cancer-targeting treatment. To our knowledge, this would be the first example of FcRn as the targeting element for cancer treatment. We expect that SPEDOX-6 will stimulate the development of new cancer therapeutics based on FcRn targeting. Interestingly, since most tumors have low levels of FcRn (18 TPM) [45], SPEDOX-6 was speculated to be highly efficacious against various cancers.

While we have demonstrated great antitumor efficacy using mouse models, there is an expected significant difference in SPEDOX-6’s PK in mice and humans, caused by altered recycling efficiency due to differential binding affinities of different FcRn (hFcRn or mFcRn)-albumin (HSA or MSA) combinations [50,51]. The binding affinity follows the order of mFcRn-HSA << hFcRn-HSA, suggesting that SPEDOX-6 would have a much longer circulatory half-life in humans than in mice due to the very weak mFcRn-HSA affinity and HSA’s poor recycling efficiency in mice. As a result, mouse cells would trap and degrade more SPEDOX-6 than normal cells in humans. This would effectively reduce the available SPEDOX-6 to xenograft tumor cells in mice, relative to tumor cells in humans. Consequently, while a good immune deficient animal tumor model to closely mimic physiological SPEDOX-6 circulation (PK) is lacking, the antitumor efficacy of SPEDOX-6 is expected to be better in humans than in the mouse tumor models, as we previously discussed [10], which could and would be confirmed by the current phase Ib/IIa human clinical trials.

The observed difference in antitumor efficacy between 15 and 17.5 mg/kg SPEDOX-6 treatment in the HT-1080 mouse model is intriguing. Our previous study used 20 mg/kg SPEDOX-6 (MTD) (4× of DOX’s MTD) in the TNBC mouse model without showing excessive toxicity [10]. However, when we used the same dose in the initial experiment of the current study, all mice lost >20% BW and later went into moribund states, indicating exceeding MTD. We therefore tested slightly reduced doses of 15 and 17.5 mg/kg, with the latter being the true MTD in the HT-1080 mouse model for the current study. While the dose difference is relatively small (17%), the antitumor efficacy differs significantly in both male and female mice. In males, the TV change over 20 days was −73% and −89% for 15 and 17.5 mg/kg treatment, respectively, while females had +24% and −84% change under the same conditions. The result indicates that the SPEDOX-6 dose–response curve is very steep for HT-1080 as a highly sensitive tumor [52], and a suboptimal dose may result in poor antitumor efficacy [53].

HT-1080 was originally isolated from connective tissue as a soft tissue sarcoma (fibrosarcoma) cell line. However, a recent study found that it has an R132C mutation in IDH1 [42], which is commonly associated with chondrosarcomas cell lines. While HT-1080 is still commonly used as an STS cell line, it has been reclassified as a dedifferentiated chondrosarcoma line for bone cancer studies. In this context, SPEDOX-6’s high antitumor efficacy against HT-1080 may offer some implication that SPEDOX-6 may be effective against some bone cancers.

## 5. Conclusions

This study, as part of IND-enabling studies, has established that the cardiotoxicity of SPEDOX-6 at a high dose of 25 mg/kg for two cycles (total amount of 50 mg/kg for 50 days) in an SD rat model is undetectable. It also demonstrated that the total exposure (AUC_last_) of SPEDOX-6 at three dose levels is 7–17 times higher than DOX at equivalent doses for both genders. In addition, this study reaffirms SPEDOX-6 as an efficacious antitumor nanodrug without increased systemic toxicity compared to DOX. GLP-grade SPEDOX-6’s MTD has been established at 30 mg/kg DOX-equivalent (>8× of free DOX’s MTD) between different tumor models. While SPEDOX-6 can achieve high antitumor efficacy on TNBC (MDA-MB-231) with a medium FcRn expression level of 18 TPM [10], it is more potent against STS (HT-1080) xenografts with the lowest FcRn expression level at <2 TPM among all cancer cell lines in the database [45]. Furthermore, it is less effective against ES (SK-ES-1) with the highest FcRn expression level at 893 TPM in the database. SPEDOX-6’s antitumor efficacy displays an apparent inverse relationship with FcRn expression levels. Therefore, SPEDOX-6 may be used to effectively target tumors with low levels of FcRn expression, offering potential clinical applications across a variety of cancer types. The current human phase Ib/IIa clinical trials under the way will provide more information on establishing the correlation between SPEDOX-6’s antitumor efficacy and FcRn expression levels of STS cancer patients. If validated by clinical trials, SPEDOX-6 will be the first targeted therapy based on FcRn as the targeting element, thereby leading to the development of new cancer-targeting therapeutics and their broad clinical applications.

## Figures and Tables

**Figure 1 cancers-17-00881-f001:**
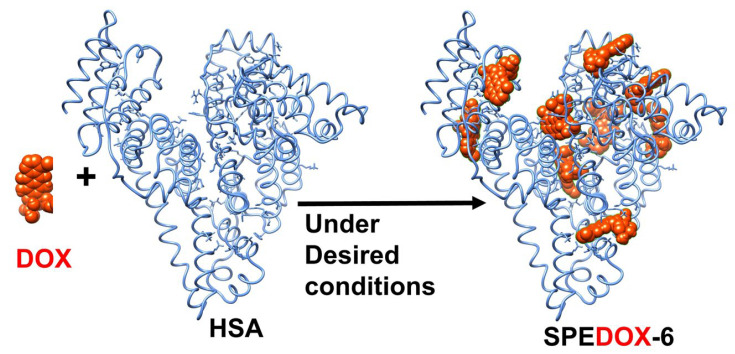
Computer docking images of SPEDOX-6 (HSA:DOX = 1:9, A_547_/A_481_ = 0.53).

**Figure 2 cancers-17-00881-f002:**
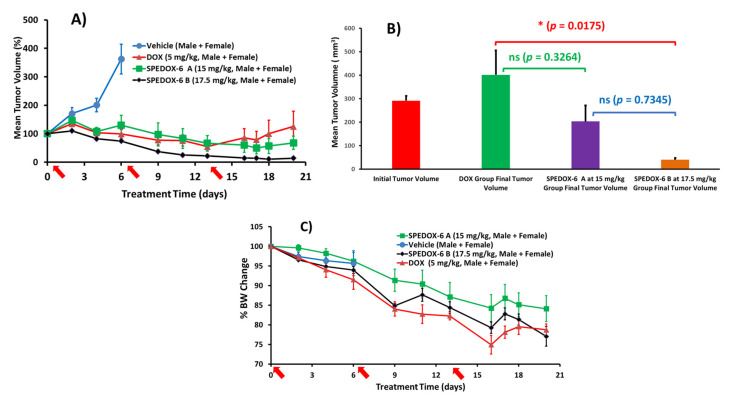
SPEDOX-6’s antitumor efficacy study in HT-1080 animal model, drug injections on Days 0, 6, and 13 indicated by arrow signs. (**A**) Mean TV vs. treatment time for all mice. Mice # for each group, control (n = 8), DOX treatment (n = 8), SPEDOX-6 A at 15 mg/kg (n = 9), and SPEDOX-6 Bat 17.5 mg/kg (n = 10). On Day 6, control group is significantly different from the 3 treatment groups (****, *p* < 0.0001); on Day 20, SPEDOX-6 B group significantly reduced TV compared to DOX group (*, *p* = 0.0175). (**B**) Comparison of the final TV for the treatment groups on Day 20. (**C**) Mean BW change vs. treatment time for all mice, not significantly different from each other.

**Figure 3 cancers-17-00881-f003:**
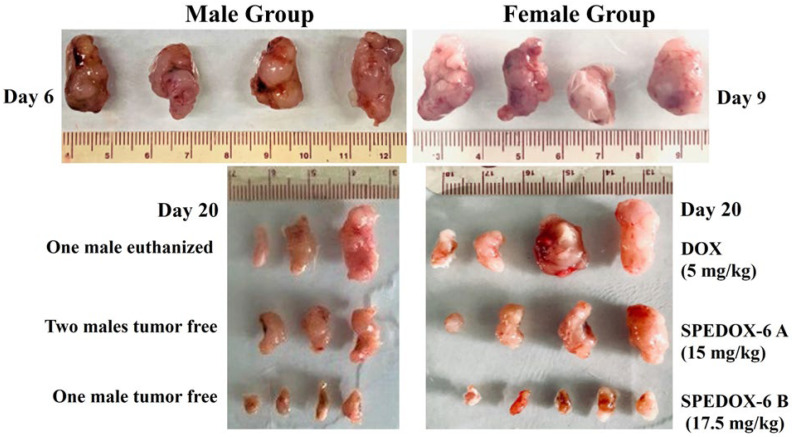
Photographic images of tumors of each group at the end of experiments. (**Top**) Control groups on Days 6 and 9 when mice were euthanized due to fast tumor growth. (**Bottom**) Treatment groups on Day 20 when the experiment ended. One male was sacrificed on Day 12 due to large tumor size. On Day 20, 2/5 and 1/5 males reached tumor-free status (complete remission).

**Figure 4 cancers-17-00881-f004:**
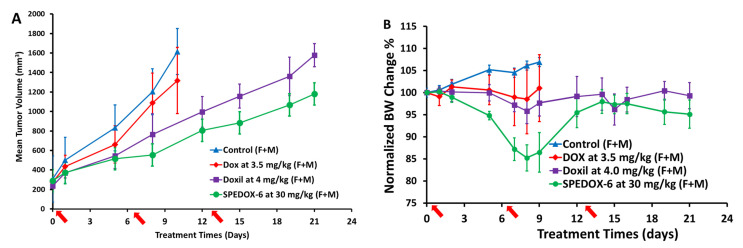
SPEDOX-6’s antitumor efficacy study in SK-ES-1 animal model drug injections at Days 0, 6, and 13, indicated by arrow signs. (**A**) Mean TV vs. treatment time for all mice. Each testing group contains 4 male and 4 female mice. On Day 8, DOX had 18.4% TGI, lower than control group (NS, *p* = 0.557); Doxil treatment reached 36.4% TGI, significantly lower than control group (*, *p* = 0.0116); SPEDOX-6 achieved 54.0% TGI, significantly lower than control group (***, *p* < 0.0001); SPEDOX-6 significantly reduced tumor volume compared to DOX group (*, *p* = 0.0183). On Day 21, SPEDOX-6 significantly reduced tumor volume compared to Doxil group (**, *p* = 0.0016), calculated by 2-way ANOVA and mixed-effects analysis. (**B**) Mean BW change vs. treatment time for all mice, not significantly different from each other (ns, *p* > 0.05).

**Figure 5 cancers-17-00881-f005:**
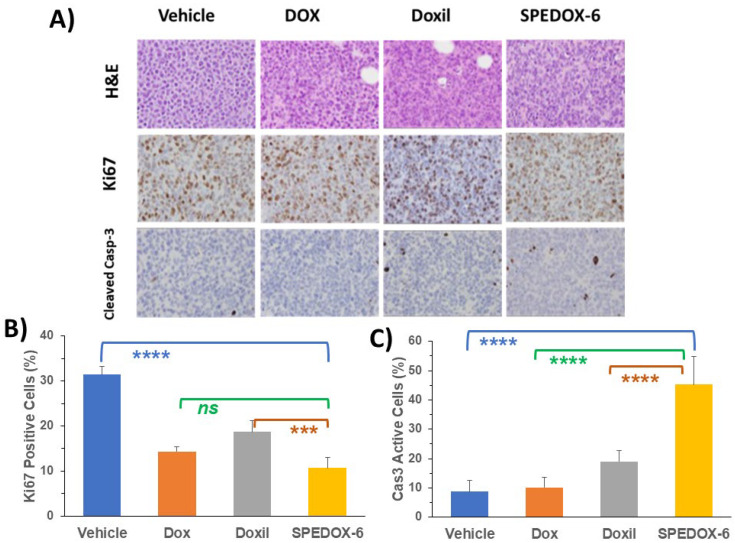
Immunohistochemical tissue following drug treatment. (**A**) Staining images (40X) of paraffin-embedded tumor tissues (SK-ES-1) sections for H&E, Ki67, and cleaved/active caspase-3 for tumor tissues for control group, DOX (3.5 mg/kg), Doxil (4.0 mg/kg), and SPEDOX-6 (30 mg/kg). (**B**) Comparison of Ki67-positive cells among different treatments. The difference is significant between SPEDOX-6 and Doxil (***, *p* = 0.001) and between SPEDOX-6 and control (****, *p* < 0.0001), but no significance is found between SPEDOX-6 and Dox (ns, *p* = 0.2827). (**C**) Comparison of caspase-3 active cells. There is significant statistical difference between SPEDOX-6 and the other 3 samples (****, *p* < 0.0001).

**Figure 6 cancers-17-00881-f006:**
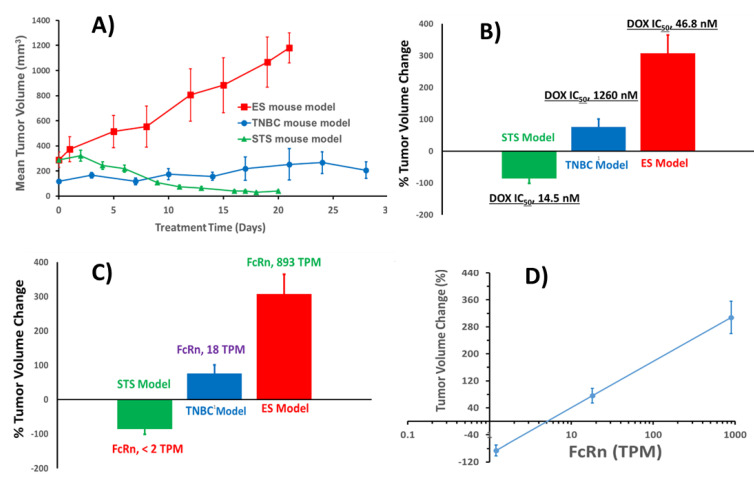
Antitumor efficacy of different tumor models in relation to DOX IC_50_ and FcRn expression levels. (**A**) Mean TV vs. treatment times for STS, TNBC, and ES mouse models treated with SPEDOX-6 at 20, 17.5, and 30 mg/kg, respectively; (**B**) TV change at end of the treatment vs. DOX IC_50_ values for STS, TNBC, and ES; (**C**) TV change at end of the treatment vs. FcRn levels of ES, TNBC, and STS mouse models; (**D**) plot of TV change at end of treatment vs. log FcRn value of the three mouse models.

**Table 1 cancers-17-00881-t001:** Semi-quantitative scoring criteria for cardiotoxicity.

Proportion and Severity of Myofibrillar Loss and Vacuolization	Score
Normal myocardial ultrastructural morphology	0
Not completely normal but no evidence of anthracycline-specific damage	0.5
Isolated myocytes affected and/or early myofibrillar loss; damage to <5% of all cells	1
Isolated myocytes affected and/or early myofibrillar loss; damage to 5–15% of all cells	1.5
Clusters of myocytes affected by myofibrillar loss and/or vacuolization, with damage to 16–25% of all cells	2
Clusters of myocytes affected by myofibrillar loss and/or vacuolization, with damage to 26–35% of all cells	2.5
Severe, diffuse myocyte damage (>35% of all cells)	3

**Table 2 cancers-17-00881-t002:** Key TK parameter of DOX in SD rats.

Compound	Day	Dose Level	Gender	t_max_	C_max_	AUC_last_
		(mg/kg)		(h)	(ng/mL)	(h*ng/mL)
DOX	Day 1	3.5	Male	0.083	2534.95	744.49
			Female	0.083	2307.74	793.90
	Day 22	3.5	Male	0.083	4198.95	1288.16
			Female	0.083	2528.94	858.69

**Table 3 cancers-17-00881-t003:** Key TK parameter of SPEDOX in SD rats.

Compound	Day	Dose Level	Gender	t_max_	C_max_	AUC_last_
		(mg/kg)		(h)	(ng/mL)	(h*ng/mL)
SPEDOX-6	Day 1	7.5 (low)	Male	0.083	26,835.92	11,542.86
			Female	0.083	28,146.48	12,602.36
		15 (middle)	Male	0.083	40,765.74	23,636.64
			Female	0.083	61,393.95	31,227.69
		25 (high)	Male	0.083	122,356.92	61,104.74
			Female	0.083	100,741.65	62,791.56
	Day 22	7.5 (low)	Male	0.083	63,685.04	28,148.54
			Female	0.083	51,579.66	23,531.06
		15 (middle)	Male	0.083	155,160.70	71,875.30
			Female	0.083	123,030.52	58,042.20
		25 (high)	Male	0.083	235,137.91	135,681.08
			Female	0.083	202,399.68	105,246.05

**Table 4 cancers-17-00881-t004:** Comparison of SPEDOX-6 with DOX for systemic exposures.

Day	SPEDOX-6/ DOX Dose Ratio	SD Male Rats			
		C_max_ Ratio	C_max_ Ratio/Dose Ratio	AUC_last_ Ratio	AUC_last_ Ratio/Dose Ratio
D1	2.14	10.59	**4.94**	15.50	**7.24**
D22	2.14	15.17	**7.08**	21.85	**10.20**
D1	4.29	16.08	**3.75**	31.75	**7.41**
D22	4.29	36.95	**8.62**	55.80	**13.02**
D1	7.14	48.27	**6.76**	82.08	**11.49**
D22	7.14	56.00	**7.84**	105.33	**14.75**
		SD Female Rats			
D1	2.14	12.20	**5.69**	15.87	**7.41**
D22	2.14	20.40	**9.52**	27.40	**12.79**
D1	4.29	26.60	**6.21**	39.33	**9.18**
D22	4.29	48.65	**11.35**	67.59	**15.77**
D1	7.14	43.65	**6.11**	79.09	**11.07**
D22	7.14	80.02	**11.20**	122.57	**17.16**

**Table 5 cancers-17-00881-t005:** Summary of antitumor efficacy of DOX and SPEDOX-6 on HT-1080.

	Control Group	DOX Group (5 mg/kg, qwk × 2)	SPEDOX-6 A Group (15 mg/kg, qwk × 3)	SPEDOX-6 B Group (17.5 mg/kg, qwk × 3)
Cumulative dose (mg/kg)	0	10	45	52.5
DOX MTD enhancement	n/a	1×	4.5×	5.25×
Ending day	6	20	20	20
Initial mean tumor volume (mm^3^)	282.5	291.5	313.69	289.2
Mean tumor volume (mm^3^) at ending day	964.3	401.4	202.93	39.8
% tumor volume change at ending day	241.3	37.7	−35.3	−86.2
Mean % BW change at ending day	95.7	79.1	84.1	77.1

**Table 6 cancers-17-00881-t006:** Summary of antitumor efficacy of DOX, Doxil, and SPEDOX-6 on SK-ES-1.

	Control Group	DOX Group (3.5 mg/kg, qwk × 3)	Doxil Group (4 mg/kg, qwk × 3)	SPEDOX-6 Group (30 mg/kg, qwk × 3)
Cumulative dose (mg/kg)	0	10.5	12	90
DOX MTD enhancement	n/a	1×	1.14×	8.57×
Ending day	10	10	21	21
Initial mean tumor volume (mm^3^)	306.5	275.4	235.8	289.6
Mean tumor volume (mm^3^) at ending day	1203.3	1313.9	1577.8	1180.1
% tumor volume change at ending day	292.6	377.1	569.1	307.4
Mean % BW change at ending day	106.9	101.1	99.3	95.1

**Table 7 cancers-17-00881-t007:** Summary of three mouse model studies on STS, TNBC, and ES.

	STS (HT-1080)	TNBC (MDA-MB-231)	ES (SK-ES-1)
**Mouse strain**	SCID	Nude BALB/c	SCID
**DOX IC_50_ (nM)**	14.5 nM	1260 nM	46.8 nM
**FcRn Level (TPM)**	1.1 TPM (<2)	18 TPM	893 TPM
**SPEDOX-6 grade**	Non-GLP	Non-GLP	GLP
**SPEDOX-6 dosing at MTD**	17.5 mg/kg	20 mg/kg	30 mg/kg
**Mean % TV change at ending day**	−86% (Reduction)	76% (Increase)	307% (Increase)

## Data Availability

Data are available on reasonable request. All data generated or analyzed during the current study are included either in this article or in the Appendix A.

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
