# Peer review of "Targeted Treatment of Sarcomas by Single Protein Encapsulated Doxorubicin with Undetectable Cardiotoxicity and Superior Efficacy"

_cancers, 2025, doi:10.3390/cancers17050881_

Round 1
Reviewer 1 Report
Comments and Suggestions for Authors
The overall goal of the study, to develop a therapy option using SPEDOX-6 that can improve treatment outcome of tumors that are usually treated with doxorubicin is a very important one. However, this reviewer believes that the study could be greatly improved by including the following points:
· Since it is mentioned in line 316 that there is a change observed for food consumption and organs (what does this specifically mean?) I expect to see the data for these two parameters in the results or supplemental information.
· In line 359/360. It is stated that 5-10mg/kg Doxo induces cardiotoxicity but in the experiments performed in this project the cardiotoxicity score is 0. Could the authors please elaborate in the text on this description in findings e.g. in number of injections, difference in accumulative dose, time of cardiotoxicity analysis post treatment, the method used to determine cardiotoxicity, species etc.?
· This reviewer believes the visualization of the animal experiments could be improved to use little cartoons depicting the experimental setup or include injections time/moments in the graphs.
· Please depict the statistics (with * or ns) in the graphs of Figure 2(A and C), 4, 6 and supplemental figures S1 till S5.
· For Fig 2A please indicate in the legend which group is SPEDOX-6 A and which group is SPEDOX-6 B as mentioned in the text.
· A statement is made about the significant differences between genders and between doses (line 432/433. This reviewer expect to see the plots (bar graphs) for TV vs 1) treatment groups, 2) gender and 3) CR/PR including statistics.
· Since significant statement are made about the data in Table 5, please include statistics in this Table.
· In line 451 – 455 the statement is made that SPEDOX-6 treatment is not displaying higher systemic toxicity compared to doxorubicin. However, in figure S5 a clear decrease in BW is observed for both male and female animals. Based on this data this reviewer would conclude also the SPEDOX-6 treatment is quite toxic to these animals (similar to doxorubicin). And this toxicity can not be explained by the tumour burden, since the tumours (especially for high SPEDOX-6) are very small.
· How can the difference in SPEDOX-6 concentration between non-GLP and GLP-grade be explained (line469)?
· In line 474 the TGI of different treatment groups is compared at day 8, way not day 10 (which is the last day Control and Doxorubicin mice are included in the experiment)?
· Figure 4: which statistical test are used in these plots, on which time points? Please specify in graph and figure legend.
· Figure 4B. For the SPEDOX-6 group a clear BW loss is observed around day 8. In line489 – 491 the authors claim no (high) toxicity is observed. This reviewer believes this claim can not be made based on the data in Fig 4B. Although these animals recover their BW this reviewer is wondering what type of toxicity these animals experienced ideally by showing pathology of (at least the heart) of these animals vs the other experimental groups.
· In line 491-493 a statement is made about gender bias. Please provide blots/bar graphs showing the data including statistics to substantiate this claim.
· Stainings in figure 5 are done on tissue taken at different time points after start/end treatment. This reviewer is doubting if the conclusion that are drawn from this data can be made (line 498-506). At least emphasize in the text what would be the influence/limitation of the timing of tissue collection related to treatment time on the conclusion that are made.
· Please score the stainings shown in figure 5 and perform statistics.
· The data described in the discussion line 559 – 592 including data showed in Table 7 and figure 6 should be moved to the results section.
· TV vs treatment is compared between the 3 different tumour models in figure 6A. Are all treatment setups the same? Please indicate in the text were treatments are different when making conclusions about these different models.
· The authors state that that for tumours with lower levels of FcRn, there is lower conversion of SPEDOX-6, and subsequent longer exposure to the drugs. This reviewer is wondering if there is therefore a higher toxicity in low FcRn compared to high FcRn tumors?
· It would be interesting to test the correlation of FcRn levels and the efficacy of SPEDOX-6 in an in vitro/cell line setting including a panel of lines with different FcRn levels.
· The authors state that SPEDOX-6 is less cardiotoxic than Doxorubicin treatment. Would this imply that SPEDOX-6 treatment could be given over a longer period of time to better control tumour grow (when there is no CR)? And could the authors test this in their animal model with only PR?
· It would be interesting to test the location and recycling of SPEDOX-6 in vitro/ cell lines. E.g. by looking by microscopy using stainings for the endolysosomal system/recycling compartments in FcRn high and low cell types.
·
· The toxicity for SPEDOX-6 is studied in rats, however, since some clear BW loss is observed for some of the mice experiments it would be interesting to check cardiotoxicity and general pathology in these mice.
· In the different tumour models different doses of the drugs are use. These differences are partially explained throughout the text, however, this reviewer would like to see the following data to better compare efficacy and toxicity:
- Measure TV at qual dose (now SPEDOX-6 is claimed to be more effective, but also much higher doses are used)
- Determine toxicity (general and cardiotoxicity) at equal effective dose.
Comments on the Quality of English LanguageTextual the manuscript could be greatly improved by correcting some textual errors, rewriting of some of the text and including extra information. In general the text would be improved to provide more background to the performed experiments by better describing why the experiments are performed and how this relate to what is already knows.
Textual/data visual suggestions:
· In the abstract there are some statements made about the FcRn status of the tumor cell lines and that SPEDOX-6 can be the first targeted therapy based on the FcRn expression level. However, there is no clear introduction about the RcRn expression level in sarcoma tumour cells or its role in the disease or why this is important for the effectivity of the SPEDOX-6. Only in the discussion FcRn expression level and mechanism of action for SPEDOX-6 in relation to FcRn is explainen. This reviewer believes the abstract and introduction could be strongly improve if this information would be added, so that reader can better grasp the importance of targeting RcRn in this tumor and what is new/important/better of your SPEDOX-6 compared to what is already available.
· Although STS and DOX where already introduced in the abstract. I would include the full name and abbreviation again when first used in the introduction.
· In the title of the supplemental information document the & symbol is used instead of “and”
· I assume a mistake was made in the text explaining the treatment scheme in the methods section (line 200 and 201)? (once every 3 weeks and 4 consecutive weeks) or please better explain the timing of actions.
· 2 x “were described” in line 222
· Line error at line 278 and 279
· 2x full stop at line 310
· Methods en results are partly mixed up, and methods are unnecessary long. Some results are described in the methods section with references to the results and supplemental information. This reviewer believes the manuscript would improve by a clear separation from results in the results section and supplemental information and a to the point description of the experiments in the methods section.
· The results section is at the moment a very dry numeration of experimental outcome without any introduction about why a specific experiment is performed / how the experiments relates to what is known or what the outcome of these experiments improves the field/therapy. (specifically section 3.1) This makes it very monotonous and uninteresting to read. The manuscript would therefor greatly improves if some context can be added to the different experiments.
· Font size line 325/326
· In line 316 it is referred to fig. S1, the next one is Fig S4 in line 326. Please make S4 à S2. Also correct the order of the rest of the references to supplemental figures. Furthermore, a clear explanation of the data in S4 is missing when there is referred to the figure.
· Table 4 is referred to before Table 3. Please make sure that all tables come in the order the manuscript is written.
· The legend of Table 4 could be improved by specifying what is compared between SPEDOX-6 and DOXO
· Figure legends of Fig. S1, S2 and S3 are missing. Please include a proper figure legend describing the data presented.
· In line 364 it is described that male rats are more sensitive to the treatment as measured in body weight loss compared to female animals. Can the authors please describe in the text how this relates to other findings on toxicity of these type of drugs in mice/rats and humans?
· The statement/conclusion in line 372-373 is unclear. Please rewrite or better specify.
· Experiment describes in line 383 – 388: is the data included in the manuscript. Than please refer to the data. Or state (data not shown).
· The title of table 5 could be more specific, e.g. by including tumour model information etc.
· A conclusion on how well SPEDOX-6 controls tumour grow in the HT-1080 is missing. There is only a reference to the summarized data in Table 5 (line 415).
· Please refer to the supplemental tables S1 -S6 in the main text, not only in the text of the supplemental text.
· Incorrect page bleak line 423
· For clarification please use the terminology of SPEDOX-6 A group and SPEDOX-6 B group (as is done in the main text/results) also in the figures.
· Please check all font sizes of figures so all axis are uniform and readable.
· Figure 4A, y-as labels misspelled.
· Figure 4A and B, use of capitals in x-as not consistent
· Line 487 “&” symbol is used in a normal sentence. Please change for “and”.
Clearly state that the information in line 579-592 is a hypothesis/theory and not yet experimental tested.
Author Response
Reviewer # 1
Point 1: Since it is mentioned in line 316 that there is a change observed for food consumption and organs (what does this specifically mean?) I expect to see the data for these two parameters in the results or supplemental information.
A: we added food consumption figures into Fig. S1 as panel C and D. Due to the large amount of data on organ mass (different mice, organs, and doses), it is difficult to plot organ mass data due to multiple organs, dose, and rat. Organ mass reductions are related to toxicity of anticancer drugs. However, the data follows the general trend observed for BW and food consumption. we revised text accordingly.
Point 2: In line 359/360. It is stated that 5-10mg/kg Doxo induces cardiotoxicity but in the experiments performed in this project the cardiotoxicity score is 0. Could the authors please elaborate in the text on this description in findings e.g. in number of injections, difference in accumulative dose, time of cardiotoxicity analysis posttreatment, the method used to determine cardiotoxicity, species.?
A: In our MS, we stated: “However, literature reported that a single injection of high dose of DOX at 5 -10 mg/kg to male SD rats has induced the cardiotoxicity (40)”. We cited the ref. # 40, a single injection of DOX at 5-10 mg/kg in SD rat model caused the cardiotoxicity, we did not perform any experiment for this study because it is our intention.
Point 3: This reviewer believes the visualization of the animal experiments could be improved to use little cartoons depicting the experimental setup or include injections time/moments in the graphs.
A: we added arrow signs to Day when drug solution was injected, as shown in in Fig. 2A, 2C, Fig. 4A, Fig. 4B, Fig.S5A, B, C, D. We also add “drug injections at day 0, 6 and 13, indicated by arrow signs” into Fig.2, Fig. 4 and Fig.S5.
Point 4: Please depict the statistics (with * or ns) in the graphs of Figure 2(A and C), 4, 6 and supplemental figures S1 till S5.
A: We have add statistics with in figures or figure legends.
Point 5: For Fig 2A please indicate in the legend which group is SPEDOX-6 A and which group is SPEDOX-6 B as mentioned in the text.
A: We have changed Fig. 2 and Fig. S5, with the desired SPEDOX-6 A and SPEDOX-6 B labels on all graphs.
Point 6: A statement is made about the significant differences between genders and between doses (line 432/433. This reviewer expect to see the plots (bar graphs) for TV vs 1) treatment groups, 2) gender and 3) CR/PR including statistics.
A: While the averaged TV difference is significance between genders between does, statistic difference between them is not (ns, p>0.05). We discuss this in the text and add p values in Fig. 2B.
Point 7: Since significant statement are made about the data in Table 5, please include statistics in this Table.
A: In Table 5, we edited “SPEDOX-6 A” and “SPEDOX-6 B” in the last two columns. Furthermore, this table is reflections of both Fig. 2A and Fig, 2C in numbers, its statistic is in Fig. 2A and Fig 2 legend.
Point 8: In line 451 – 455 the statement is made that SPEDOX-6 treatment is not displaying higher systemic toxicity compared to doxorubicin. However, in figure S5 a clear decrease in BW is observed for both male and female animals. Based on this data this reviewer would conclude also the SPEDOX-6 treatment is quite toxic to these animals (similar to doxorubicin). And this toxicity cannot be explained by the tumour burden, since the tumours (especially for high SPEDOX-6) are very small.
A: In DOX group, total 10 mg/kg (2 x 5) of DOX after 2 injections was used. In SPEDOX-6 B group, total 52.5 mg/kg (3 x 17.5) DOX equivalent after 3 injections was used, meaning that 52.5/10 = 5.25 times of DOX equivalent has been used, when comparing DOX group to SPEDOX-6 B group. However, Fig. 2C, Fig. S5B and Fig. S5D indicated that % BW loss is very similar for both DOX group and SPEDOX-6 B group, not significantly different. Yes, SPEDOX-6 is toxic to those animals with 5.25 times higher dosing amount of DOX equivalent, which has achieved the significant reductions of TV in comparison to DOX group while maintaining similar toxicity as DOX. This is the ultimate goal for developing novel anticancer drugs, the highest antitumor efficacy and the lowest toxicity. In fact, all chemo drugs are quite toxic.
Point 9: How can the difference in SPEDOX-6 concentration between non-GLP and GLP-grade be explained (line 469)?
A: Non-GLP SPEDOX-6 was prepared using the lab equipment. GLP-grade SPEDOX-6 was prepared by an FDA-certified GMP-manufacturer under the compliance of GLP. GLP-grade SPEDOX-6 usually have much better quality because the manufacturing processes were highly controlled and optimized, and lyophilizing system under GLP/GMP compliance for making SPEDOX-6 cake/power has the precise temperature control, which can substantially reduce formation of impurities and free DOX in SPEDOX-6. HPLC analysis using size exclusion column for this batch of GLP-grade SPEDOX-6 (Lot #: R1059-01-091) has shown that free DOX is very low at 0.73%. In the GMP batch of SPEDOX-6 (Lot #: 23SD015), free DOX is <0.5%. Due to low amount of free DOX in GLP –grade SPEDOX-6, GLP-grader SPEDOX-6 would be much less toxic, which can explain why almost 2X GLP-grade SPEDOX-6 can be used in SK-ES-1 mouse model study.
Point 10: In line 474 the TGI of different treatment groups is compared at day 8, way not day 10 (which is the last day Control and Doxorubicin mice are included in the experiment)?
A: Typically, TV measurements are done twice per week. However, mice in both control and DOX groups had to be euthanized due to tumor burdens after measuring their TVs. But for mice at Doxil and SPEDOX-6 groups, their TVs were not measured on Day 10, therefore, TGI couldn’t be calculated, only TGI at day 8 can be calculated and shown in Fig. 4.
Point 11: Figure 4: which statistical test are used in these plots, on which time points? Please specify in graph and figure legend.
A: In legend of Fig 4A, we have clearly shown times points of statistical test, at Day 8 and Day 21. In the experimental section, we provided the detailed statistic test, for Fig. 4, we used “2 way ANOVA and Mixed-effects analysis”, in order to clarify, we added a paragraph in Fig. 4A legend.
Point 12: Figure 4B. For the SPEDOX-6 group a clear BW loss is observed around day 8. In line 489 – 491 the authors claim no (high) toxicity is observed. This reviewer believes this claim can not be made based on the data in Fig 4B. Although these animals recover their BW this reviewer is wondering what type of toxicity these animals experienced ideally by showing pathology of (at least the heart) of these animals vs the other experimental groups.
A: In Fig. 4B, the dose of SPEDOX-6 is 30 mg/kg of DOX equivalent, which is 8.57 times higher than DOX at 3.5 mg/kg. In this case, mouse % BW reduction for SPEDOX-8 groups were observed at early time points, < 80 %, which is acceptable according to NCI’s guidance. The almost completely recovery of % BW changes at late time points indicated that toxicity of SPEDOX-6 is reversible, which can be used for justifying SPEDOX-6’s toxicity. In the previous section, line 354 to line 359, we have studied cardiotoxicity of SD rats treated by dose up to 2x25 mg/kg (total doses of 50 mg/kg) of SPEDOX-6 and longer time (50 days), there is undetectable cardiotoxicity (the score was 0 based on the scoring criteria). We speculate that mice in SPEDOX-6 group at Fig. 4B should not develop cardiotoxicity that is irreversible. Since % BW change as one of drug toxicity indicators, it is hard to pin down what types of toxicity.
Point 13. In line 491-493 a statement is made about gender bias. Please provide blots/bar graphs showing the data including statistics to substantiate this claim.
A: Line 491 to line 493, we didn’t state any gender bias. WE stated “By carefully analyzing antitumor efficacy and toxicity data for 4 study groups in males and females, no gender bias in the mouse model with SK-ES-1 was observed”.
Point 14, Staining in figure 5 are done on tissue taken at different time points after start/end treatment. This reviewer is doubting if the conclusion that are drawn from this data can be made (line 498- 506). At least emphasize in the text what would be the influence/limitation of the timing of tissue collection related to treatment time on the conclusion that are made.
Please score the staining shown in figure 5 and perform statistics.
A: We have analyzed Ki67- and caspase 3-staining of all samples. Two new plots (Fig 5B and 5C) have been added, along with statistics.
Point 15, The data described in the discussion line 559 – 592 including data showed in Table 7 and figure 6 should be moved to the results section.
A: Table 7 and Fig. 6 have been moved to Results Section, along with their associated text.
Point 16, TV vs treatment is compared between the 3 different tumour models in figure 6A. Are all treatment setups the same? Please indicate in the text were treatments are different when making conclusions about these different models.
A: In Table 7, we have listed all differences in three-mouse model study, for examples, for MDA-MB-231, mouse strain: Nude BALB/c, for HT1080 and SK-ES-1, mouse strain: SCID. For MDA-MB-231 and SK-ES-1, SPEDOX-6 grade, Non-GLP, for SK-ES-1, SPEDOX-6 grade, GLP. Therefore, we have stated the different setups in Fig. 6A.
Point 17: The authors state that that for tumours with lower levels of FcRn, there is lower conversion of SPEDOX-6, and subsequent longer exposure to the drugs. This reviewer is wondering if there is therefore a higher toxicity in low FcRn compared to high FcRn tumors?
A: This is an excellent point raised by the reviewer. In line 587 to line 588, we stated that “Low concentration of FcRn reduces SPEDOX-6 recycling, leading to increased SPEDOX-6 accumulation in the tumor cell.” We didn’t state “for tumours with lower levels of FcRn, there is lower conversion of SPEDOX-6, and subsequent longer exposure to the drugs”. We don’t have direct evidence to speculate any relationship between FcRn levels and the toxicity of SPEDOX-6. However, cellular toxicity is caused by DOX in its free form, which can be taken up by healthy cells/tissues/organs such as heart. Since SPEDOX-6 effectively encapsulates DOX, leaving <1% free DOX, the toxicity of SPEDOX-6 is low, compared with DOX. Low FcRn levels in tumor cells reduce SPEDOX-6 recycling, thereby trapping and increasing SPEDOX-6 concentration in tumor cells, which leads to increased free DOX concentration in tumor cells following its dissociation from HSA and/or enzymatic degradation of HSA. However, FcRn levels in tumor cells don’t impact FcRn levels in other healthy cells. SPEDOX-6 recycling in healthy cells is not affected by FcRn levels in tumor cells. Therefore, we don’t expect that tumor FcRn levels would increase SPEDOX-6’s side effect toxicity. We have added this briefly in discussion.
Point 18: It would be interesting to test the correlation of FcRn levels and the efficacy of SPEDOX-6 in an in vitro/cell line setting including a panel of lines with different FcRn levels.
We thank the reviewer for the thoughtful comment. We have tested a number of human cancer cell lines with different levels of FcRn levels. Cell proliferation assays showed that there were no significant correlation between the SPEDOX-6’s inhibitory effects and FcRn levels in these cell lines. Although the preliminary in vitro results were not consistent with our hypothesis, we think several confounding factors may mask the effect of FcRn on SPEDOX-6 action. 1) Different gene mutation profiles (e.g. p53 and PI3K pathway mutations) in these cell lines affect cancer cell survival capacity in SPEDOX-6 treatment. 2) The in vitro culture condition, which leads to rapid cell proliferation, do not faithfully recapitulate the in vivo tumor tissue microenvironment. We found that SPEDOX-6 is internalized into cells within 1-2 hours after being added to the culture. It is known that cancer cells degrade albumin as energy and nutrition sources. In vitro growing cells may degrade SPEDOX-6 rapidly. To systematically address whether FcRn affects the SPEDOX-6 effect in vitro, we plan to test an expanded cell line panel with well defined gene mutation profiles and use reduced serum culture conditions in future studies.
Point 19: The authors state that SPEDOX-6 is less cardiotoxic than Doxorubicin treatment. Would this imply that SPEDOX-6 treatment could be given over a longer period of time to better control tumour grow (when there is no CR)? And could the authors test this in their animal model with only PR?
A: Since SPEDOX-6 is less cardiotoxic than DOX, there are two treatment options, (1) multi-fold higher dose of SPEDOX-6 when same number of treatment cycles are used; (2) the same dose of SPEDOX-6 when more cycles are used. In current study, we tested the first treatment option because higher antitumor efficacy by multi-fold SPEDOX-6 might overcome the acquired drug resistance of DOX due to low dose. The longer treatment by SPEDOX-6 at lower doses may be undesirable because the acquired DOX resistance may have substantially reduced SPEDOX-6’s antitumor efficacy. We may test it in our current human clinical trial on cancer patients who were previously treated by the standard DOX agent.
Point 20. It would be interesting to test the location and recycling of SPEDOX-6 in vitro/ cell lines. E.g. by looking by microscopy using stainings for the endolysosomal system/recycling compartments in FcRn high and low cell types.
We appreciate the reviewer’s constructive comment. In our recent paper on SPEDOX-6 (PMID: 37657447. Arzt et al), immunostaining showed that SPEDOX-6, in a similar manner as human serum albumin was able to internalize within cancer cells within hours of treatment. Both nuclear and cytoplasmic accumulatio of SPEDOX-6 were detected in three cancer cell models. Of note, previous studies have shown that human serum albumin can be detected in cytoplasmic and nuclear fractions of cancer cells. Our ongoing studies will address whether SPEDOX-6 are co-localized with lysosome markers and whether SPEDOX-6 levels in lysosomes are inversely correlated with FcRn levels.
Point 21: The toxicity for SPEDOX-6 is studied in rats, however, since some clear BW loss is observed for some of the mice experiments it would be interesting to check cardiotoxicity and general pathology in these mice.
A: In our GLP-toxicology study, we have systematically studied TK and pathology of >200 SD rats at different doses of SPEDOX-6 with negative and positive control (DOX), because the FDA required rat model that is much closer to the human than mouse model. In our IND applications, approved by US FDA, we have demonstrated the undetectable cardiotoxicity of those SD rats up to 50 days. Therefore, it is not necessary to test the mouse model. Furthermore, we will carefully monitor the human heart functions when conducting human clinical trials of SPEDOX-6 for treating soft tissue sarcomas.
Point 22: In the different tumour models different doses of the drugs are use. These differences are partially explained throughout the text, however, this reviewer would like to see the following data to better compare efficacy and toxicity:
- Measure TV at qual dose (now SPEDOX-6 is claimed to be more effective, but also much higher doses are used)
A: SPEDOX-6 at low dose is ineffective in mice due to its effective encapsulation. However different doses (from 25 mg/kg up to 310 mg/kg) will be evaluated in our current human phase Ib/IIa clinical trials. The trial will therefore address the question.
- Determine toxicity (general and cardiotoxicity) at equal effective dose.
A: In our GLP-toxicology study, we used three doses of SPEDOX-6 for study, in comparison to DOX at 3.5 mg/kg as a positive control. At low dose (7.5 mg/kg) of SPEDOX-6, which is 2.14 time higher than DOX, this treatment group caused much less BW loss in male and female (Fig. S4A and Fig. S4B) and had much higher food consumptions in male and female (Fig. S4C and Fig. S4D). Clearly, in this SD rat model, 2.14 times higher dose of SPEDOX-6 is much less toxic, compared to DOX. In pathology report, both DOX at 3.5 mg/kg and SPEDOX-6 at 7.5 mg/kg did not cause any cardiotoxicity.
In Fig. 2C, SPEDOX-6 at 15 mg/kg (total 45 mg/kg) has lower % BW loss in comparison to DOX at 5 mg/kg (total 10 mg/kg), while SPEDOX-6 showed higher anticancer efficacy compared to DOX.
Based on the toxic data from two rodent models, it is clear that SPEDOX-6 has much lower toxicity at the same dose in comparison to DOX.
Point 23: In the abstract there are some statements made about the FcRn status of the tumor cell lines and that SPEDOX-6 can be the first targeted therapy based on the FcRn expression level. However, there is no clear introduction about the RcRn expression level in sarcoma tumour cells or its role in the disease or why this is important for the effectivity of the SPEDOX-6. Only in the discussion FcRn expression level and mechanism of action for SPEDOX-6 in relation to FcRn is explained. This reviewer believes the abstract and introduction could be strongly improve if this information would be added, so that reader can better grasp the importance of targeting RcRn in this tumor and what is new/important/better of your SPEDOX-6 compared to what is already available.
We thank the reviewer for the suggestion. We have added FcRn related statements in both abstract and introduction. More statements on FcRn are in Discussion.
Point 24: Although STS and DOX where already introduced in the abstract. I would include the full name and abbreviation again when first used in the introduction.
The suggestion is implemented into the text.
Point 25: In the title of the supplemental information document the &
symbol is used instead of “and”
A: It has been changed.
Point 26: I assume a mistake was made in the text explaining the treatment scheme in the methods section (line 200 and 201)? (once every 3 weeks and 4 consecutive weeks) or please better explain the timing of actions.
A: “once every 3 weeks and 4 consecutive weeks” is the standard GLP-toxicology terminology, it means: The drug will be injected into animal every 3 weeks in total 4 weeks, which indicated that the animals will have two injections and be euthanized after total 4 weeks.
Point 27: 2 x “were described” in line 222
Corrected.
Point 28. Line error at line 278 and 279
Corrected.
Point 29: 2x full stop at line 310
Corrected.
Point 30: Methods en results are partly mixed up, and methods are unnecessary long. Some results are described in the methods section with references to the results and supplemental information. This reviewer believes the manuscript would improve by a clear separation from results in the results section and supplemental information and a to the point description of the experiments in the methods section.
A: Thank you for your constructive suggestion. We have substantially reorganized the text, moving all data (Figs, Tables) to Results. Fig # is also changed to be consistent with text. The renamed Table 1 lists criteria for toxicity and therefore remains in Methods.
Point 31: The results section is at the moment a very dry numeration of experimental outcome without any introduction about why a specific experiment is performed / how the experiments relates to what is known or what the outcome of these experiments improves the field/therapy. (specifically section 3.1) This makes it very monotonous and uninteresting to read. The manuscript would therefore greatly improves if some context can be added to the different experiments.
A: We have taken up your excellent suggestions. We added a paragraph right after Results. Additional text is introduced in appropriate locations to make reading more smooth and enjoyable.
Point 32: Font size line 325/326
Corrected.
Point 33: In line 316 it is referred to fig. S1, the next one is Fig S4 in line 326. Please make S4 à S2. Also correct the order of the rest of the references to supplemental figures. Furthermore, a clear explanation of the data in S4 is missing when there is referred to the figure.
A: We have reorganized fig, table #.
Point 34: Table 4 is referred to before Table 3. Please make sure that all tables come in the order the manuscript is written.
A: We have reorganized table # and they consistent now.
Point 35: The legend of Table 4 could be improved by specifying what is compared between SPEDOX-6 and DOX.
A: We have revised it as “Table 4. Comparison of SPEDOX-6 with DOX for systemic exposures”.
Point 36: Figure legends of Fig. S1, S2 and S3 are missing. Please include a proper figure legend describing the data presented.
A: All figure legends have been restored.
Point 37: In line 364 it is described that male rats are more sensitive to the treatment as measured in body weight loss compared to female animals. Can the authors please describe in the text how this relates to other findings on toxicity of these type of drugs in mice/rats and humans?
A: In our GLP-toxicology study, each group composed of total 10 SD rats (5 males and 5 females). Based on the limited numbers of SD rats, males seem to be more sensitive to the treatment, but it is not significantly (p > 0.05), unless large numbers of SD rats are used for further testing. There is no literature to report the gender bias of DOX for treating animals and human.
Point 38: The statement/conclusion in line 372-373 is unclear. Please rewrite or better specify.
A: We have revised the sentence into “the GLP-grade SPEDOX-6 at the highest dose (25 mg/kg) has increased DOX amount by 7.14 times in comparison to DOX as positive control at 3.5 mg/kg in SD rat model.”
Point 39: · Experiment describes in line 383 – 388: is the data included in the manuscript. Then please refer to the data. Or state (data not shown).
A: In those sentences, we just described what we have observed on the toxicity from SPEDOX-6 at 20 mg/kg (ref. cited), and then we reduced the dose of SPEDOX-6 into 17.5 mg/kg. In our opinion, it seems clear.
Point 40: The title of table 5 could be more specific, e.g. by including tumour model information etc.
A: We have added HT-1080 to the title.
Point 41: A conclusion on how well SPEDOX-6 controls tumour grow in the HT-1080 is missing. There is only a reference to the summarized data in Table 5 (line 415).
The referred text location is in BW section. A few lines above the section we stated “SPEDOX-6 A & B groups shrunk tumor size by 35% and 86%, respectively.”
Point 42: Please refer to the supplemental tables S1 -S6 in the main text, not only in the text of the supplemental text.
A: We have added text for each of cited fig and tables.
Point 43: Incorrect page bleak line 423.
A: Corrected.
Reviewer 2 Report
Comments and Suggestions for Authors
The manuscript «Targeted Treatment of Soft Tissue Sarcomas by Single Protein Encapsulated Doxorubicin with Undetectable Cardiotoxicity and Superior Efficacy» by Changjun Yu et al. is dedicated to the newly developed single-protein encapsulated DOX nanopreparation, SPEDOX-6, which provides a new approach to increase the effectiveness of DOX and reduce its side effects.
The manuscript is written in a scientific style, it presents the results obtained using modern methods, which makes it attractive to readers of the journal "Cancers".
The manuscript can be published in the journal "Cancers" after revision:
- The authors should correct the disadvantage in the entire text of the manuscript, which causes difficulties in understanding the article. Abbreviations must be disclosed when they are first mentioned in the text.
- In my opinion, the authors neglected to describe the relevance of their research, insufficiently revealing the fact that anthracycline antibiotics, in particular doxorubicin, are currently widely used as the basis for the creation of antitumor agents. This information would help the authors to emphasize that the use of doxorubicin in research of this kind is at the forefront of creating unique anticancer drugs!
In this regard, in the second paragraph of the "Introduction" section, the authors should provide information on the results of experimental work describing the widespread use of anthracycline derivatives. I could recommend to the authors (at their discretion) to mention the works of recent years:
- Plotnikova E, Abramova O, Ostroverkhov P, Vinokurova A, Medvedev D, Tihonov S, Usachev M, Shelyagina A, Efremenko A, Feofanov A, Pankratov A, Shegay P, Grin M, Kaprin A. Conjugate of Natural Bacteriochlorin with Doxorubicin for Combined Photodynamic and Chemotherapy. Int J Mol Sci. 2024 Jun 29;25(13):7210. doi: 10.3390/ijms25137210.
- Su L, Rezaei S, Mejia G, Pandey P, Dit Fouque KJ, Fernandez-Lima F, McGoron A, He J, Leng F. Conjugating Daunorubicin and Doxorubicin to GTP by Formaldehyde to Overcome Drug Resistance. ChemMedChem. 2024 Dec 2;19(23):e202300481. doi: 10.1002/cmdc.202300481. Epub 2024 Oct 8. PMID: 39136598; PMCID: PMC11620955.
- Yoon, H., Savoy, E. A., Mesbahi, N., Hendricksen, A. T., March, G. L., Fulton, M. D., Backer, B. S., & Berkman, C. E. (2024). A PSMA-targeted doxorubicin small-molecule drug conjugate. Bioorganic & medicinal chemistry letters, 104, 129712. https://doi.org/10.1016/j.bmcl.2024.129712.
- Kalashnikova AA, Toibazarova AB, Artyushin OI, Anikina LV, Globa AA, Klemenkova ZS, Andreev MV, Radchenko EV, Palyulin VA, Aleksandrova YR, et al. Design of New Daunorubicin Derivatives with High Cytotoxic Potential. International Journal of Molecular Sciences. 2025; 26(3):1270. https://doi.org/10.3390/ijms26031270.
- Jin C, Li J, Yang X, Zhou S, Li C, Yu J, Wang Z, Wang D, He Z, Jiang Y, Wang Y. Doxorubicin-isoniazid conjugate regulates immune response and tumor microenvironment to enhance cancer therapy. Int J Pharm. 2023 Jan 25;631:122509. doi: 10.1016/j.ijpharm.2022.122509.
- One of the key focuses of this manuscript is the modulation of Fcr expression. However, in my opinion, the authors paid very limited attention to FcRn. The authors should expand the information about FcRn in the "Discussion" section.
- In general, the authors need to design the manuscript in accordance with the rules of the MDPI journals, since currently the manuscript is carelessly designed for a number of parameters.
In conclusion, I would like to thank the team of authors for their research and their desire to present it to the scientific community, as well as wish them success in their future work.
Author Response
Reviewer # 2
The manuscript can be published in the journal "Cancers" after revision:
- The authors should correct the disadvantage in the entire text of the manuscript, which causes difficulties in understanding the article. Abbreviations must be disclosed when they are first mentioned in the text.
A: We have taken the advice, reorganize the MS and added text, and added full term before abbreviations. We believe the MS is greatly improved.
- In my opinion, the authors neglected to describe the relevance of their research, insufficiently revealing the fact that anthracycline antibiotics, in particular doxorubicin, are currently widely used as the basis for the creation of antitumor agents. This information would help the authors to emphasize that the use of doxorubicin in research of this kind is at the forefront of creating unique anticancer drugs! In this regard, in the second paragraph of the "Introduction" section, the authors should provide information on the results of experimental work describing the widespread use of anthracycline derivatives. I could recommend to the authors (at their discretion) to mention the works of recent years:
- Plotnikova E, Abramova O, Ostroverkhov P, Vinokurova A, Medvedev D, Tihonov S, Usachev M, Shelyagina A, Efremenko A, Feofanov A, Pankratov A, Shegay P, Grin M, Kaprin A. Conjugate of Natural Bacteriochlorin with Doxorubicin for Combined Photodynamic and Chemotherapy. Int J Mol Sci. 2024 Jun 29;25(13):7210. doi: 10.3390/ijms25137210.
- Su L, Rezaei S, Mejia G, Pandey P, Dit Fouque KJ, Fernandez Lima F, McGoron A, He J, Leng F. Conjugating Daunorubicin and Doxorubicin to GTP by Formaldehyde to Overcome Drug   Resistance. ChemMedChem. 2024 Dec 2;19(23):e202300481. doi: 10.1002/cmdc.202300481. Epub 2024 Oct 8. PMID: 39136598; PMCID: PMC11620955.
- Yoon, H., Savoy, E. A., Mesbahi, N., Hendricksen, A. T., March, G. L., Fulton, M. D., Backer, B. S., & Berkman, C. E. (2024). A PSMA targeted doxorubicin small-molecule drug conjugate. Bioorganic & medicinal chemistry letters, 104, 129712. https://doi.org/10.1016/j.bmcl.2024.129712.
- Kalashnikova AA, Toibazarova AB, Artyushin OI, Anikina LV, Globa AA, Klemenkova ZS, Andreev MV, Radchenko EV, Palyulin VA, Aleksandrova YR, et al. Design of New Daunorubicin Derivatives with High Cytotoxic Potential. International Journal of Molecular Sciences. 2025; 26(3):1270. https://doi.org/10.3390/ijms26031270.
- Jin C, Li J, Yang X, Zhou S, Li C, Yu J, Wang Z, Wang D, He Z, Jiang Y, Wang Y. Doxorubicin-isoniazid conjugate regulates immune response and tumor microenvironment to enhance cancer therapy. Int J Pharm. 2023 Jan 25;631:122509. doi: 10.1016/j.ijpharm.2022.122509.
A: We agree with the reviewer and have added 3 references and associated text to reflect the importance of other anthracycline.
- One of the key focuses of this manuscript is the modulation of Fcr expression. However, in my opinion, the authors paid very limited attention to FcRn. The authors should expand the information about FcRn in the "Discussion" section.
A: We agree with the reviewer, we have added FcRn related text in Introduction, Results, and Discussion.
- In general, the authors need to design the manuscript in accordance with the rules of the MDPI journals, since currently the manuscript is carelessly designed for a number of parameters. In conclusion, I would like to thank the team of authors for their research and their desire to present it to the scientific community, as well as wish them success in their future work.
A: We have formulated the MS that is consistent with the publication style. We thank the reviewer for the constructive comments and suggestions
Round 2
Reviewer 1 Report
Comments and Suggestions for Authors
Thank you for the implementation of the suggestions made in the manuscript and elaboration on questions in the response to the reviewer. This reviewer is interested to find out the results from the phase Ib/IIa clinical trail when available.